# Cytolinker Gas2L1 regulates axon morphology through microtubule-modulated actin stabilization

Dieudonnée van de Willige[1,†], Jessica JA Hummel[1,†], Celine Alkemade[2,3,†], Olga I Kahn[1], Franco KC Au[4], Robert Z Qi[4], Marileen Dogterom[2], Gijsje H Koenderink[3,‡,*] (iD), Casper C Hoogenraad[1,**] (iD) & Anna Akhmanova[1,***] (iD)

## Abstract

Crosstalk between the actin and microtubule cytoskeletons underlies cellular morphogenesis. Interactions between actin filaments and microtubules are particularly important for establishing the complex polarized morphology of neurons. Here, we characterized the neuronal function of growth arrest-specific 2-like 1 (Gas2L1), a protein that can directly bind to actin, microtubules and microtubule plus-end-tracking end binding proteins. We found that Gas2L1 promotes axon branching, but restricts axon elongation in cultured rat hippocampal neurons. Using pull-down experiments and *in vitro* reconstitution assays, in which purified Gas2L1 was combined with actin and dynamic microtubules, we demonstrated that Gas2L1 is autoinhibited. This autoinhibition is relieved by simultaneous binding to actin filaments and microtubules. In neurons, Gas2L1 primarily localizes to the actin cytoskeleton and functions as an actin stabilizer. The microtubule-binding tail region of Gas2L1 directs its actin-stabilizing activity towards the axon. We propose that Gas2L1 acts as an actin regulator, the function of which is spatially modulated by microtubules.

**Keywords** axon; cytolinker; cytoskeleton; *in vitro* reconstitution; neuronal development

**Subject Categories** Cell Adhesion, Polarity & Cytoskeleton; Neuroscience

## Introduction

The cytoskeleton is a key player in cellular morphogenesis, as it provides cells with structural support and acts as a scaffold for organelle positioning. An example of a process where cytoskeletal filaments play intricate roles is neuronal development. Neurons have complex morphologies that allow them to form elaborate networks and propagate signals in the brain. Developing neurons undergo extensive cell shape changes, which are coordinated by guidance cues relayed to the actin and microtubule (MT) cytoskeletons (reviewed in Ref. [1]).

In particular, the crosstalk between MTs and actin plays an essential role during axon maturation (reviewed in Refs [2–4]). At the tips of axonal processes, specialized structures called growth cones determine the direction and rate of axon advance. Growth cones contain a central dynamic MT array which probes the actin-rich periphery. Axon outgrowth is preceded by MT stabilization in filopodia at the tip of the growth cone, and conversely, repellent cues restrict peripheral MT entry. In a similar process, axon branching is believed to start with the formation of an actin patch along the axon, either *de novo* or as a remnant of a pausing growth cone (reviewed in Refs [5,6]). At the site of branch formation, newly generated dynamic MT plus ends are stabilized on the actin patch to initiate a new branch.

Actin-MT crosslinking proteins, also referred to as cytolinkers, are obvious candidates to regulate cytoskeletal crosstalk during axon development (reviewed in Ref. [7]). Most studies in this context have focussed on spectraplakins, a family of large proteins, which directly bind both to actin filaments and to MT shafts and indirectly associate with growing MT plus ends through end binding (EB) proteins [8–11]. The actin-MT crosslinking abilities of ACF7 (a spectraplakin otherwise known as MACF1) and its *Drosophila* ortholog Short stop (Shot) are necessary for axon extension [12,13]. Other MT plus end-associated proteins, such as CLASP and APC, participate in regulating axon outgrowth, possibly also by coordinating actin-MT coupling [14,15].

Gas2L1 (growth arrest-specific 2-like 1) is a much smaller cytolinker with a domain composition similar to ACF7, but its role is less well understood. Like ACF7, Gas2L1 contains an N-terminal

1 Department of Biology, Cell Biology, Faculty of Science, Utrecht University, Utrecht, The Netherlands
2 Department of Bionanoscience, Kavli Institute of Nanoscience, Delft University of Technology, Delft, The Netherlands
3 Living Matter Department, AMOLF, Amsterdam, The Netherlands
4 Division of Life Science and State Key Laboratory of Molecular Neuroscience, The Hong Kong University of Science and Technology, Hong Kong, China
*Corresponding author. Tel: +31 015 27 89806; E-mail: g.koenderink@amolf.nl
**Corresponding author. Tel: +31 30 2534585; E-mail: c.hoogenraad@uu.nl
***Corresponding author. Tel: +31 302532328; E-mail: a.akhmanova@uu.nl
†These authors contributed equally to this work
‡Present address: Department of Bionanoscience, Kavli Institute of Nanoscience, Delft University of Technology, Delft, The Netherlands

actin-binding calponin homology (CH) domain, a MT lattice-binding Gas2-related (GAR) domain and a C-terminal SxIP motif, which mediates the interaction with MT plus ends via EB proteins [16–18]. The actin-MT crosslinking abilities of Gas2L1 and its *Drosophila* ortholog Pigs (Pickled eggs) have been previously demonstrated in cells [18,19]. So far, Gas2L1 was shown to regulate the distance between centrioles in cycling cells [20], and Pigs was identified as a cytoskeletal target of Notch signalling, which participates in *Drosophila* wing muscle development and oogenesis [21]. However, Gas2L1 has not been studied in the context of neuronal development, although Gas2L1 mRNA is abundant in mammalian brain tissue [16]. Here, we reveal that Gas2L1 participates in regulating axon outgrowth and branching in developing mammalian neurons. By combining data obtained in primary rat hippocampal neurons, *in vitro* reconstitution assays and biochemical experiments, we show that Gas2L1 is autoinhibited and requires the simultaneous binding of both actin filaments and MTs to fully relieve this autoinhibition. Our data suggest that Gas2L1 locally regulates the actin cytoskeleton during axon maturation by stabilizing actin in response to MT binding. Gas2L1 hereby promotes axon branching while tempering axon extension.

## Results

### Gas2L1 affects axon branching and outgrowth in developing neurons

To determine whether Gas2L1 plays a role in neuronal development, we examined the effects of Gas2L1 depletion and overexpression on axon development in dissociated primary rat hippocampal neurons. Depletion resulted in a ~ 64% reduction of Gas2L1 mRNA as determined by qPCR, for which day *in vitro* (DIV) 0 neurons were electroporated with the empty vector or Gas2L1 shRNA-encoding plasmids and subjected to puromycin selection before mRNA isolation at DIV3 (Figs 1A, and EV1A and B).

The depletion of Gas2L1 from DIV0 to DIV3 resulted in less complex axons (Fig 1B) characterized by an increased axon branch length and reduced branch density (the number of branches per unit of axon length; Fig 1C–F). We note that in this set of experiments, total axon length was slightly reduced, whereas other experiments showed no significant difference (Figs EV1E and 5G). The effects of Gas2L1 depletion on neuronal morphology could be rescued by co-expressing low levels of Gas2L1 (Figs 1H and I, and EV1E–G), indicating that this phenotype is specific. Importantly, neuronal polarity was not affected by the loss of Gas2L1, as evidenced by staining of the axon initial segment marker TRIM46 in neurons depleted of Gas2L1 (Fig 1B) [22].

When Gas2L1 was overexpressed from DIV0 to DIV3, the length of the primary axon and axonal branches was decreased, whereas the branch density was increased (Fig 1C–F). These axon maturation phenotypes were opposite to those seen after Gas2L1 depletion (Fig 1C–F). Gas2L1 overexpression also led to enlargement of axonal growth cones (Fig 1G). We also noted the development of excessive filopodia (Fig EV1C), as evidenced by their co-localization with a filopodia marker fascin (Fig 1J). In addition, Gas2L1 neurons formed structures that resembled lamellipodia (Fig EV1C) and displayed some punctate staining with the lamellipodia markers

cortactin and p34-Arc/ARPC2 (Fig EV1H and I). We note that these features were beyond the level of detail included in our morphological analyses, which were based on axon tracings (Fig EV1D). Despite the obvious morphological defects induced by overexpression, axons of Gas2L1-overexpressing neurons were still polarized, as TRIM46 staining appeared normal in these neurons (Fig 1B).

We conclude that Gas2L1 stimulates formation of new axon branches but restricts their elongation. The latter effect is different from that of ACF7, a structurally related cytolinker that promotes both axon outgrowth and branching [12,13]. The ability of a protein to interact with both actin filaments and MTs can thus be associated with distinct neurodevelopmental functions.

### Gas2L1 specifically localizes to actin-MT overlaps in *in vitro* reconstitution assays

To better understand how Gas2L1 mediates actin-MT crosstalk, we purified full-length Gas2L1, its actin-binding CH domain and its MT-binding C-terminal fragment, which was termed Tail (Figs 2A and EV2A). We studied the behaviour of these proteins in an *in vitro* reconstitution assay, which included dynamic MTs and actin filaments stabilized with phalloidin [23]. We immobilized stable MT seeds on a glass surface, allowing dynamic MTs to grow in a solution containing tubulin dimers and unattached free-moving actin filaments (Fig 2B).

As the first step, we tested the behaviour of Gas2L1 and mutants in the presence of dynamic MTs or stabilized actin filaments alone. Full-length Gas2L1 did not bind to single actin filaments in the absence of MTs (Fig 2C), whereas the CH domain did (Figs 2D and EV2B). Similarly, in an assay with MTs alone, full-length Gas2L1 did not bind MTs (Fig 2E). By contrast, the Tail domain was able to bind MTs in the absence of actin filaments (Fig 2F). Moreover, Gas2L1 did not track growing MT plus ends in the presence of EB3, whereas the Tail fragment did (Fig EV2C and D). These results indicate that under the tested conditions, individual domains of Gas2L1 are capable of binding actin filaments, the MT lattice and MT plus ends, but the full-length protein is not.

Surprisingly, when we added Gas2L1 to a composite assay with both MTs and low concentrations of actin filaments, we observed that Gas2L1 exclusively accumulated at the sites where actin filaments and MTs overlapped (Fig 2G). Specific accumulation of Gas2L1 on actin-MT overlaps was especially obvious from events during which an actin filament gradually aligned with a growing MT: the appearance of the Gas2L1 signal along the MT coincided with the zippering of the unbound part of an actin filament along the MT (Fig 2G). Similar zippering of actin filaments and MTs was observed previously with an engineered cytolinker containing a CH domain targeting actin and an SxIP motif targeting EB proteins [24]. In a composite assay where more actin filaments were available for binding, we observed MTs covered with multiple co-aligned actin filaments (Fig 2H), whereas no MT-actin co-alignment occurred without Gas2L1 (Fig EV2H).

Time lapse analysis of actin-MT-Gas2L1 bundles revealed that MTs inside these bundles remained dynamic (Fig 2I and J). When a catastrophe occurred and the MT shrunk back, Gas2L1 did not remain bound to actin, but disappeared together with the MT (Fig 2I and J), and the actin filaments that were initially co-aligned with the MT dispersed (Movie EV1, Fig 2K). Once the MT repolymerized, the

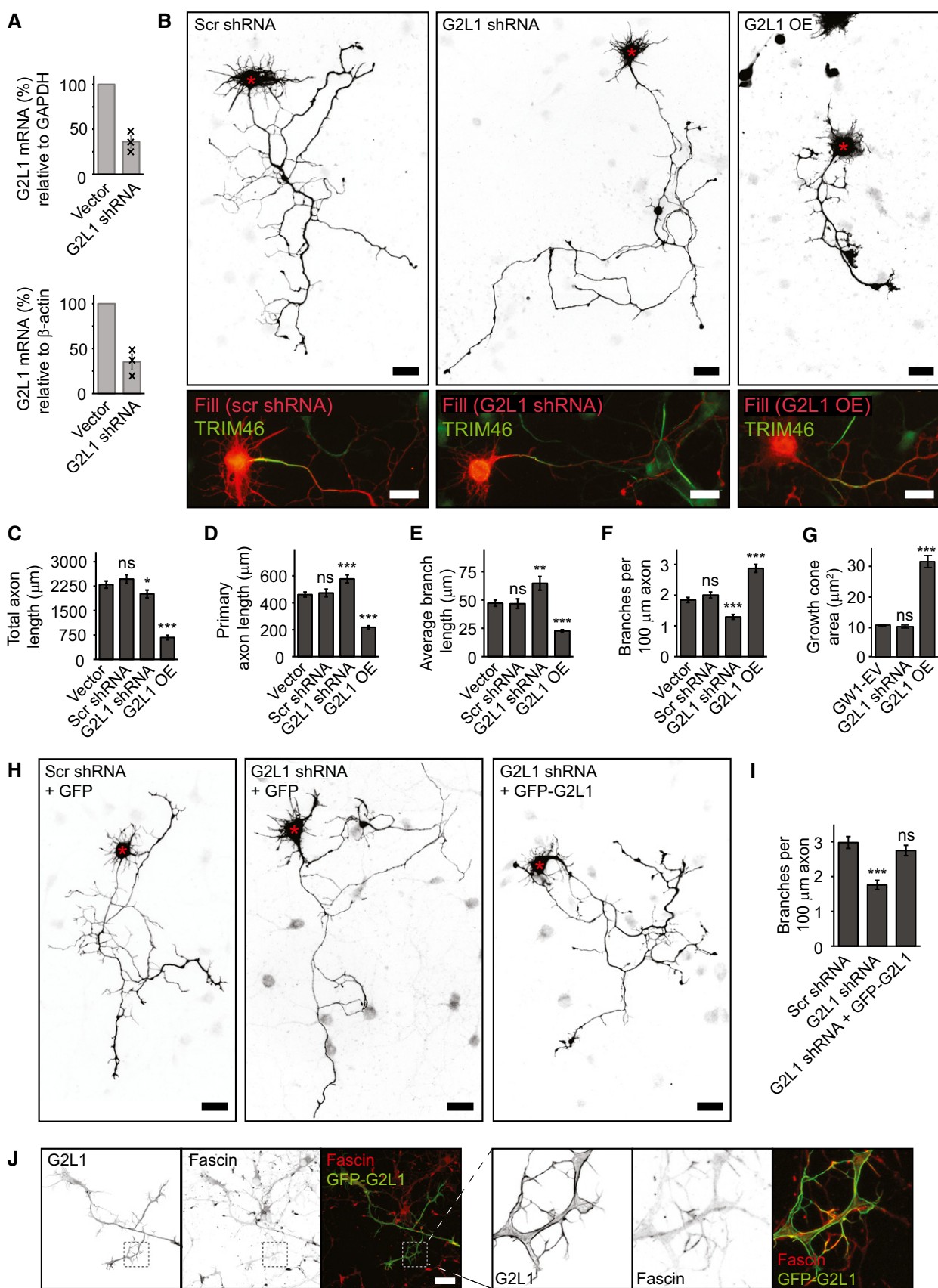

**Figure 1.**

◄

**Figure 1.  Gas2L1 balances axon outgrowth and branching in developing neurons.**

A    qPCR experiments showing the decrease in Gas2L1 (G2L1) mRNA levels upon DIV0-DIV3 shRNA treatment of primary rat hippocampal neurons (A; $n$ = 3 biological replicates, black crosses represent individual data points). Electroporated neurons were subjected to 48-h puromycin selection prior to mRNA isolation.

B    Silhouettes (composite images from β-galactosidase fill, top panels) of DIV3 neurons treated with scrambled (Scr) or G2L1 shRNA and co-expressing HA-β-galactosidase, or overexpressing HA-G2L1 and HA-β-galactosidase for 3 days. Red asterisks indicate the position of the soma. Bottom panels show fill (red) combined with TRIM46 staining (green) of the neurons shown in the panels above.

C–F    Quantifications of total axon length (C), primary axon length (D), average (non-primary) branch length (E) and the number of branches per 100 μm axon (F) for DIV3 neurons treated as described in (B). Vector = empty pSuper shRNA vector co-expressing HA-β-galactosidase. $n$ = 50 neurons per condition from three independent experiments.

G    Quantification of the growth cone area in DIV4 neurons transfected with GW1-EV (empty GW1 vector), G2L1 shRNA or HA-G2L1 and HA-β-galactosidase for 1 day. $n$ = 130–159 growth cones from 37 to 43 neurons per condition (159 growth cones from 43 neurons for GW1-EV, 130 growth cones from 37 neurons for G2L1 shRNA, 142 growth cones from 41 neurons for HA-G2L1) from three independent experiments.

H, I    Rescue experiments showing the number of branches per 100 μm axon in neurons co-expressing scrambled or G2L1 shRNA with GFP, or G2L1 shRNA with GFP-G2L1, and HA-β-galactosidase from DIV0 to DIV3. Silhouettes (composite images from β-galactosidase fill) are shown in (H). Red asterisks indicate the position of the soma. $n$ = 34–43 neurons per condition (36 for Scr shRNA, 34 for G2L1 shRNA, 43 for G2L1 shRNA + GFP-G2L1) from two independent experiments.

J    Localization of GFP-Gas2L1 (G2L1) in a DIV3 neuron stained for fascin (α-fascin), as well as merged images showing co-localization between GFP-Gas2L1 and fascin (left panels). Boxes indicate zoomed regions (right panels).

Data information: Scale bars: 30 μm in (B, H, J). Data are displayed as means ± SEM. Mann–Whitney test, ns: not significant, *$P$ < 0.05, **$P$ < 0.01, ***$P$ < 0.001.

accumulation of Gas2L1 and actin along its shaft was restored (Fig 2K). The addition of EB3 to the assay did not alter Gas2L1-induced MT-actin co-alignment and did not induce any enrichment of Gas2L1 at growing MT plus ends, in spite of the fact that EB3 was able to track MT plus ends in the same assay (Fig EV2E–G). These data confirm that Gas2L1 does not behave as a canonical plus-end-tracking protein although it does contain an EB-binding SxIP motif.

Our results confirm that Gas2L1 can crosslink actin filaments and MTs. However, while individual fragments of Gas2L1 are able to bind MTs or actin filaments, full-length Gas2L1 does not localize to either of these separate cytoskeletal components *in vitro*. Instead, Gas2L1 binds to MTs and actin filaments simultaneously, suggesting that the protein might be autoinhibited and that the interaction with both actin and MTs is required to relieve autoinhibition. Of note, Gas2L1 did not bind independently to MTs regardless of incubation time. However, over time Gas2L1 accumulated on slowly forming actin bundles (but never individual filaments) in the absence of MTs (Fig EV2I). Since there appears to be some interaction between Gas2L1 and actin in the absence of MTs, we propose that the binding of Gas2L1 to actin acts as the first step towards initiating actin-MT crosslinking (Fig 2L).

## Autoinhibition of Gas2L1 is a result of the interaction between the CH domain and MT-binding tail

If Gas2L1 is indeed autoinhibited, its MT-binding tail should directly compete with actin for binding its CH domain. A similar intramolecular interaction was reported for Shot: its N-terminal tandem CH domains interact with the C-terminal EF-hand/GAR region [25].

We mapped potential intramolecular interactions of Gas2L1 by testing whether its different deletion mutants (Fig 2A) interacted with the CH domain in a pull-down assay. Gas2L1 fragments were tagged with an N-terminal AviTag fused to GFP (bioGFP), which was biotinylated by co-expressing the biotin ligase BirA. Streptavidin beads were used to bind biotinylated Gas2L1 fragments and incubated with HA-tagged CH domains, which in case of interaction were retained on the beads after washing and could be detected by Western blotting.

Full-length Gas2L1 and the MT-binding Tail fragment pulled down the CH domain, whereas the biotinylated GFP, used as a negative control, displayed no binding (Fig 3A). Moreover, a Tail mutant harbouring a mutated SxIP motif (Tail-SxAA), which was previously shown to abolish Gas2L1-EB interaction [20], still bound to the CH domain. When the Tail domain was split into smaller fragments, the binding to the CH domain was lost (Fig 3A). These data suggest that the interaction of the CH domain of Gas2L1 with the MT-binding C-terminal part of the protein requires both the GAR domain and the unstructured region, and occurs independently from the association with EB proteins.

Interestingly, consistent with the pattern of *in vitro* reconstitution experiments, the CH domain of Gas2L1 pulled down endogenous actin much more efficiently than full-length Gas2L1 (Fig 3B). However, full-length Gas2L1 acquired the ability to efficiently pull-down actin in the presence of a co-expressed CH domain (Fig 3A and B). This was likely due to the additional CH domain competing with the tail part of the full-length Gas2L1 for binding to the CH domain located within the same molecule (Fig 3C), confirming our model of Gas2L1 autoinhibition. Furthermore, the MT-binding Tail fragment did not pull-down actin even though the CH domain co-precipitated (Fig 3A). This reinforces the idea that the CH domain of Gas2L1 cannot bind to actin when it is associated with the tail part of the protein in the autoinhibited conformation.

Considering the previously identified association between the CH domain and GAR/EF-hand region of Shot [25], we tested whether an interaction between the CH and GAR domains of Gas2L1 was sufficient to inhibit binding of the CH domain to actin. Indeed, we did not observe actin being pulled down by a Gas2L1 truncation mutant containing the CH and GAR domains but lacking the unstructured C-terminal region (amino acids 1–304; Fig 3D and E). This suggests that the 1–304 mutant is autoinhibited via a direct association between the GAR and CH domains. However, we could not detect the CH-GAR interaction when the CH and GAR domains were expressed independently and not tethered to each other within the same molecule (Fig 3A). This suggests that the interaction between the CH and GAR domain is rather weak and that it is likely strengthened by the presence of the unstructured C-terminal region (amino acids 305–681). Taken together, we conclude that Gas2L1 is autoinhibited through a direct interaction between its CH domain and its MT-binding tail. Our *in vitro* reconstitution observations predict that this mechanism could promote the selective activity of the protein at actin-MT interfaces.

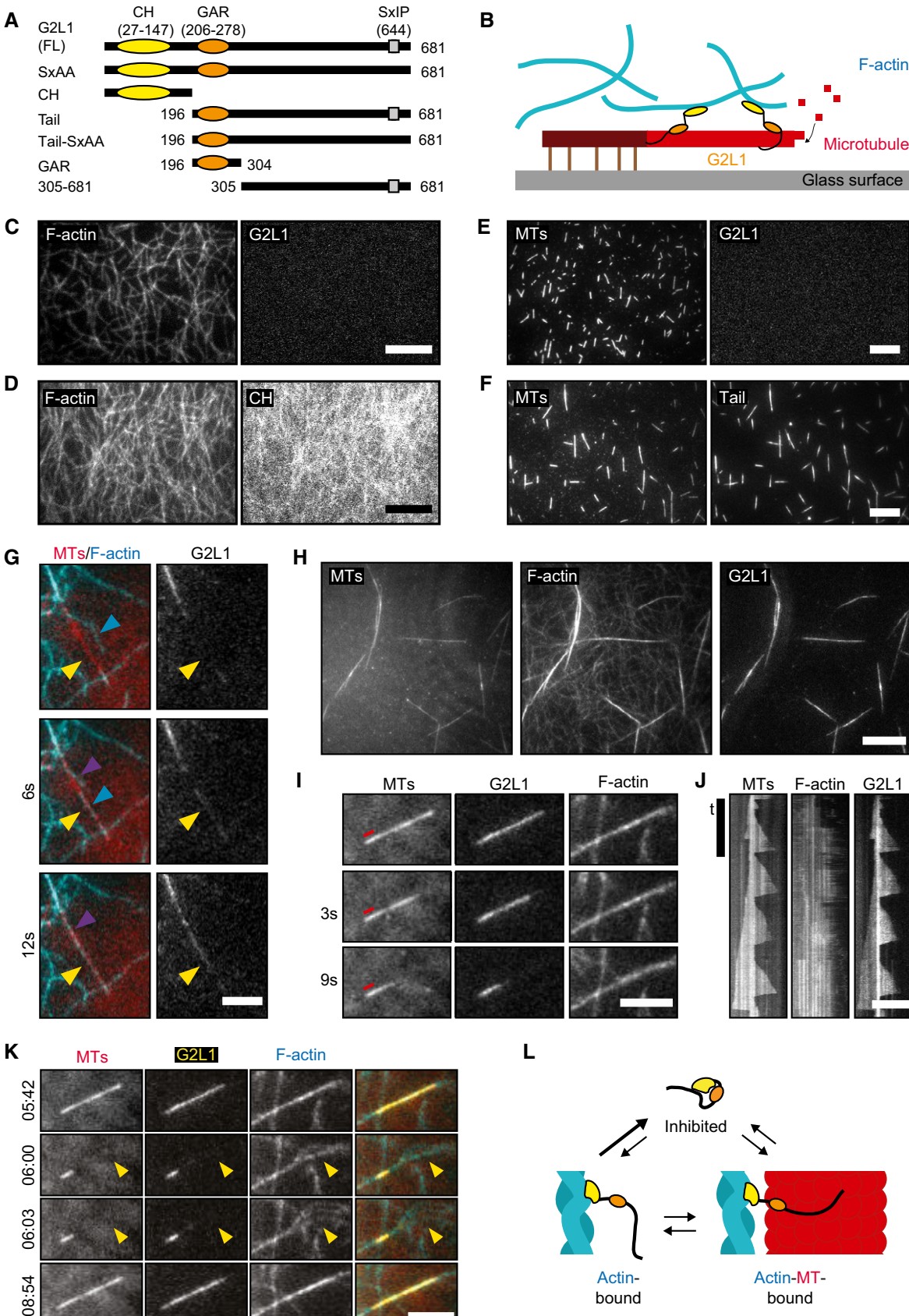

**Figure 2.**

**Figure 2.** *In vitro* reconstitution of the interaction of Gas2L1 with actin filaments and MTs.

A   Schematic depiction of the domain structure of Gas2L1 (G2L1) and Gas2L1 mutants used in this study.
B   Schematic depiction of *in vitro* TIRF assays. Stable MT seeds are attached to the glass surface, from which dynamic MTs grow by addition of tubulin dimers (either with or without EB3 present). Actin filaments are free to move around. G2L1 links F-actin to MTs.
C   *In vitro* reconstitution of full-length Gas2L1 in the presence of actin (0.5 μM) only.
D   *In vitro* reconstitution of the Gas2L1 CH domain in the presence of actin (0.5 μM) only. The intensity range between the F-actin panels and Gas2L1/CH panels of (C) and (D) are equal.
E   *In vitro* reconstitution of full-length Gas2L1 in the presence of MTs only.
F   *In vitro* reconstitution of the Gas2L1 Tail domain in the presence of MTs only.
G   *In vitro* reconstitution of full-length Gas2L1 in a composite assay with both actin (10 nM) and MTs. Gas2L1 only localizes at MT-actin overlaps. Gas2L1 accumulation (yellow arrows) occurs after an actin filament "lands" (blue arrows) and zippers (purple arrows) onto a MT.
H   *In vitro* reconstitution of full-length Gas2L1 in a composite assay as in (G), but with higher actin concentration (1 μM).
I   Still frames showing the dynamics of a Gas2L1-F-actin-MT bundle, from a composite *in vitro* reconstitution assay as described in (H). The red line indicates the (stable) MT seed.
J   Kymographs showing the dynamics of the Gas2L1-F-actin-MT bundle seen in (I).
K   Still frames taken from Movie EV1 at the indicated timestamps, showing the dispersion of actin filaments (yellow arrowheads) after Gas2L1 and the MT disappear from the Gas2L1-F-actin-MT bundle and renewed actin bundling and Gas2L1 accumulation upon MT regrowth. Experimental conditions are as described for (H).
L   Model of Gas2L1 autoinhibition.

Data information: Scale bars: 10 μm (C–F, H, I), 5 μm (G, K). For (J), vertical scale bar: 3 min, horizontal scale bar 5 μm.

## The Gas2L1 Tail fragment contains at least two MT-binding regions

Since the unstructured C-terminus of Gas2L1 promoted the autoinhibitory interaction between the GAR and CH domains, we next tested whether this part of Gas2L1 also affects the binding of the GAR domain to MTs. We investigated MT binding of the GFP-tagged Gas2L1 Tail fragment (amino acids 196–681) or its parts (Fig 2A) in COS-7 cells, where individual MTs are easier to distinguish than in neurons (Fig EV3). At very high expression levels, both the GAR domain and the unstructured C-terminus could bind to and bundle MTs, and the same was true for the complete Tail fragment (Fig EV3). However, at low overexpression levels, the GAR domain showed comparatively little association with MTs, and the unstructured C-terminus displayed a comet-like distribution characteristic for plus-end tracking proteins, whereas the complete Tail fragment co-localized with MTs (Fig EV3). Mutating the SxIP motif to SxAA, and therefore abolishing the binding of the Tail fragment to EB proteins and MT plus ends, did not significantly alter the binding of the Tail to MT shafts. These data suggest that both the unstructured C-terminus and the GAR domain of Gas2L1 have some affinity for the MT lattice and cooperate in associating with MTs. Our results match those reported for comparable truncation mutants of the *Drosophila* Gas2L ortholog Pigs [19].

## Gas2L1 localizes to neuronal F-actin and regulates axon development by influencing actin stability

Next, we turned back to primary hippocampal neurons to investigate how Gas2L1 affects their cytoskeleton. In NIH3T3 and COS7 cells, Gas2L1 was shown to localize predominantly to actin stress fibres [16,18]. In neurons, which lack stress fibres, GFP-tagged Gas2L1 primarily co-localized with the total F-actin population (Fig 4A). The localization of Gas2L1 to F-actin was also apparent in subcellular areas devoid of MTs. This observation reveals that in contrast to our *in vitro* experiments, Gas2L1 can localize to actin structures independently of MTs in neurons. This result suggests that additional layers of regulation are likely present in cells (see Discussion), and further strengthens the idea that actin binding may

be the first step towards stably relieving the autoinhibition of Gas2L1. Some Gas2L1 accumulation along MTs was also observed (Fig 4B). Interestingly, axons of DIV3-4 neurons expressing high levels of Gas2L1 showed clear cytoskeletal abnormalities: MTs frequently buckled inside growth cones (Fig 4C–E), which were enlarged due to what appeared to be increased amounts of F-actin (Figs 1G and 4C).

Given the dominant localization of Gas2L1 to F-actin, we then focussed on actin-related effects of the protein. We noticed that Gas2L1 depletion and overexpression phenotypes during axon development line up with results published in various studies investigating the effect of mild changes in actin stability on this process [26–31]. Therefore, we tested the effect of low nanomolar doses of Latrunculin B (LatB) on the primary hippocampal neurons in our culture system. DMSO, which was used as a vehicle, had no effects on axon morphology at the highest concentration used in these experiments (Fig EV4A–E). In contrast, 48 h of treatment with 10–100 nM LatB reduced branch density and induced branch elongation at DIV3 (Fig 5A–E). Mild actin destabilization with LatB thus resembled the effects of Gas2L1 knockdown (summarized in Fig 7A). However, primary axon length did not increase after LatB treatment (resulting in a net decrease in combined axon length as seen in Fig 5B and C), whereas the depletion of Gas2L1 did induce lengthening of the primary axon (Fig 1D). These data suggest that Gas2L1 could act by stabilizing actin and display higher activity in the primary axon compared to elsewhere in the neuron, possibly due to the presence of distinct actin filament populations.

F-actin stabilization by low nanomolar doses of jasplakinolide mimicked effects of Gas2L1 overexpression on axons of control vector-expressing DIV4 neurons treated for 48 h (Fig 5F–J, grey bars, and summarized in Fig 7A). Moreover, at 25 nM of jasplakinolide, axons displayed thickening and lamellipodia- and filopodia-like protrusions akin to Gas2L1 overexpression (Fig EV1C and Fig 5F). Importantly, 5–10 nM jasplakinolide was able to rescue the effects of Gas2L1 depletion on axon morphology (Fig 5F–J, white bars). Taken together, these data suggest that Gas2L1 mediates axon branching and outgrowth by increasing F-actin stability. These findings are consistent with the observation that the *Drosophila*

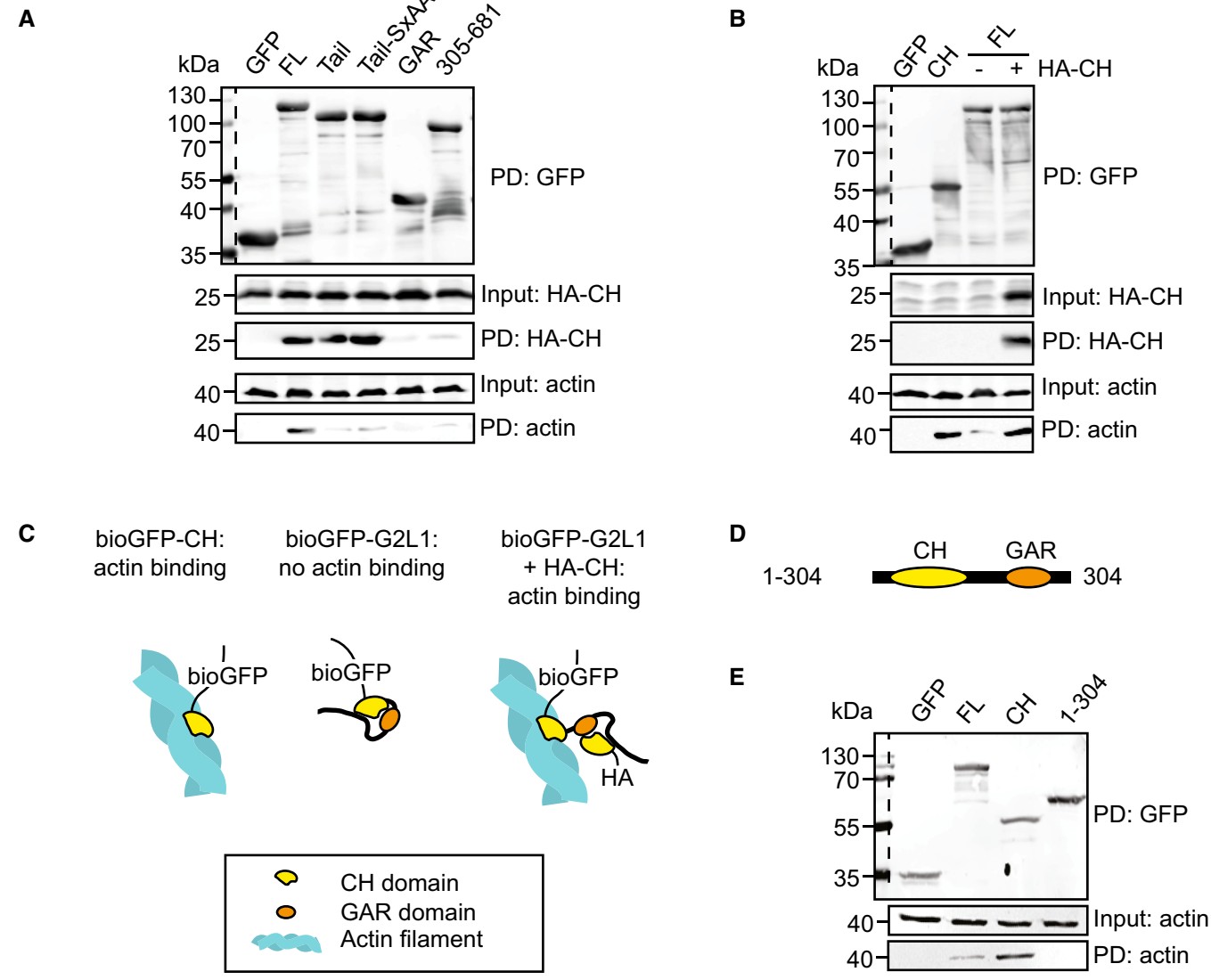

**Figure 3. Intramolecular interaction between Gas2L1 domains.**

A  Pull-down experiment showing the binding of Gas2L1's CH domain (HA-CH) to full-length Gas2L1 (FL) and both Tail and Tail-SxAA mutants in HEK293 cell lysates, as well as co-precipitation of actin with full-length Gas2L1 bound to HA-CH.

B  Pull-down experiment showing co-precipitation of actin with Gas2L1's CH domain in HEK293 cell lysates and full-length Gas2L1 only when HA-CH is co-expressed.

C  Model of increased co-precipitation of actin with Gas2L1 in the presence of HA-CH.

D  Schematic depiction of the domain structure of Gas2L1_1-304.

E  Pull-down experiment showing the co-precipitation of actin with Gas2L1's CH domain (CH), but not with full-length Gas2L1 (FL) or Gas2L1-1-304 (1–304) in HEK293 cell lysates.

Data information: Dotted lines separate marker lanes from sample lanes on the same blots (top panels). For bottom panel blots, markers were detected on the same blot, but in a different channel than the one used for signal detection that are shown here.

Gas2L-ortholog Pigs stabilizes F-actin when overexpressed in S2R+ cells [19].

### MT behaviour appears unaffected in neurons depleted of Gas2L1

We next determined whether and how MT-related effects contribute to the role of Gas2L1 in axon development. In line with reports in NIH3T3 cells [18], overexpression of Gas2L1 caused the displacement of endogenous EB1 away from MT plus ends and onto Gas2L1-decorated structures (Fig 6A), which we showed above to colocalize with actin. We confirmed that this behaviour depended on the SxIP motif of Gas2L1, as mutating the SxIP motif to SxAA abolished EB1 displacement (Fig 6B). These results are in agreement with our observation that Gas2L1 is not a plus-end tracking protein *in vitro*. However, Gas2L1 does not recruit EB3 *in vitro* (Fig EV2G), suggesting that Gas2L1-dependent EB re-localization may require additional factors or post-translational modifications absent in the assays using purified proteins. The function of the

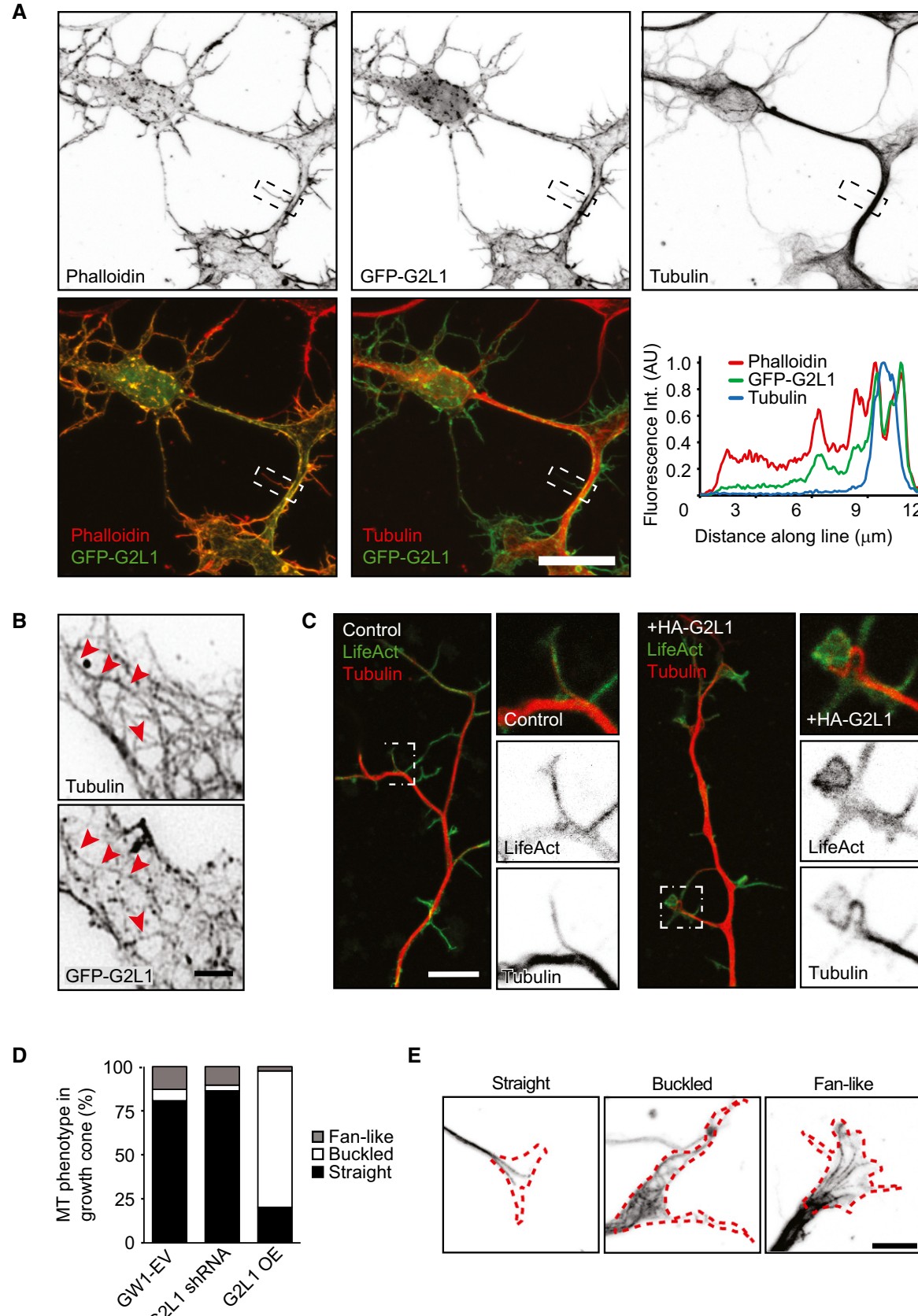

**Figure 4.**

◄

**Figure 4. Neuronal localization of Gas2L1 and effects of its overexpression.**

A   Localization of GFP-Gas2L1 (G2L1) in a DIV4 neuron stained for MTs (α-tubulin) and F-actin (Phalloidin), as well as merged images showing co-localization between GFP-Gas2L1 and Phalloidin or GFP-Gas2L1 and α-tubulin. The line scan shows co-localization along the boxed neurite.
B   Close-up of GFP-Gas2L1 localization to MTs (tubulin staining). Red arrowheads point to examples of MTs positive for GFP-Gas2L1.
C   Cytoskeletons of DIV3 neurons electroporated at DIV0, co-expressing LifeAct-GFP and control empty vector (left) or HA-Gas2L1 (right) and stained for MTs (tubulin). Boxes indicate zoomed region.
D   Quantification of MT configuration in axonal growth cones of DIV4 neurons transfected with GW1-EV (empty vector), G2L1 shRNA or HA-G2L1 (G2L1 OE) and HA-β-galactosidase for 1 day. MT configuration is classified as straight, buckled or fan-like. n = 128–214 growth cones from 30 to 36 neurons per condition (154 growth cones from 36 neurons for GW1-EV, 128 growth cones from 30 neurons for G2L1 shRNA, 214 growth cones from 36 neurons for G2L1 OE) from three independent experiments.
E   Examples of different MT configurations (from α-tubulin staining) in growth cones as quantified in (D). Dotted red line shows the growth cone silhouette (from β-galactosidase fill).

Data information: Scale bars: 20 μm in (A), 3 μm in (B), 10 μm in (C), 5 μm in (E).

EB-Gas2L1 interaction might be to augment the intrinsically weak MT affinity of Gas2L1. Of note, the redistribution of EB proteins and axon development defects induced by Gas2L1 overexpression show that, although overexpression experiments provide valuable insights into the functioning of the protein, endogenous Gas2L1 operates at low levels with more nuanced outcomes.

To determine the contribution of Gas2L1 to MT regulation at endogenous levels, we looked for signs of MT disruption in neurons depleted of Gas2L1. In cultured cell lines, overexpression of Gas2L1 or its tail fragments was reported to promote resistance to nocodazole-induced MT depolymerization and to decrease MT dynamics [16,18]. We used a photoactivatable fusion of GFP-α-tubulin to measure the longevity of MTs in axon shafts and found that although Gas2L1 overexpression could indeed increase MT lifetimes, there was no difference between control and Gas2L1-depleted neurons (Fig 6C). Similarly, we found that whereas Gas2L1 overexpression increased the ratio of acetylated/tyrosinated MTs, there was no difference between control neurons and neurons depleted of Gas2L1 (Fig EV5A and B). An increase in the ratio of acetylated/tyrosinated MTs reflects their enhanced stability, as newly polymerized MTs are tyrosinated, while older MTs become acetylated and detyrosinated [32,33]. There was no change in the growth velocity of EB3-positive MT plus ends in growth cones upon Gas2L1 depletion either (Fig 6D), and we saw no differences in MT organization in axonal growth cones of Gas2L1-depleted neurons (Fig 4D and E). These results suggest that although Gas2L1 has the potential to stabilize MTs at medium to high expression levels, MT behaviour is relatively normal in Gas2L1 knockdown neurons, likely due to the low levels at which we assume Gas2L1 to be expressed.

**MTs guide Gas2L1 activity in neurons**

Interestingly, we found that while the Gas2L1 MT-binding Tail mutant could decorate all neuronal MTs, it was enriched on TRIM46-positive MTs in proximal axons (Fig 6E). Given our *in vitro* findings which suggest a model of full release of Gas2L1 autoinhibition in the presence of actin and MTs, we wondered whether this increase in MT affinity could impact Gas2L1 activity locally. More specifically, we wondered whether the preferential interaction of the tail of Gas2L1 with MTs in the proximal axon could drive differential effects of Gas2L1 on actin stabilization.

Quantifications of the total amount of F-actin in proximal axons and non-axonal neurites (future dendrites; dendritic maturation

occurs at later stages of development) revealed that overexpression of GFP-Gas2L1, GFP-Gas2L1-SxAA and GFP-CH increased F-actin levels compared to GFP overexpression (Fig 6G and H, white bars). To determine whether the enhanced levels of F-actin also reflected enhanced actin stability, as suggested by drug-induced effects including rescue experiments (Fig 5F–J), we treated neurons overexpressing Gas2L1 fusions with high concentrations of LatB to reveal stable F-actin populations (Fig 6F–H, grey bars). Treatment of control GFP-expressing neurons with 10 μM LatB for 30 min resulted in a near-complete loss of F-actin in all projections. By contrast, some F-actin remained in the presence of overexpressed Gas2L1 fusions, although the patterns of stabilization were strikingly different between full-length Gas2L1 and mutants (Fig 6F).

To quantify and illustrate the differences in actin stabilization compared to the localization of the various fusion proteins tested, we compared the polarity indices of the localization of each Gas2L1 fusion before treatment and of F-actin distribution with or without LatB treatment (Fig 6I). The polarity indices are derived from raw fluorescence intensities in axons and non-axonal neurites (see Materials and Methods for details) and range from 1 to −1 for purely axonal or purely non-axonal localizations, respectively. A value of 0 signifies no particular enrichment. Our analysis showed that the CH domain of Gas2L1 is enriched in non-axonal neurites, where it also stabilizes actin. Surprisingly, full-length Gas2L1 was evenly distributed between the axons and other neurites, yet it most efficiently stabilized actin in the axon (Fig 6I). This suggests that the MT-binding tail region promotes axonal localization of Gas2L1 in addition to the non-axonal neurite targeting of the protein via the CH domain. Importantly, the pattern of actin stabilization by Gas2L1 mirrored the preference of the Gas2L1 tail for axonal MTs rather than the broad localization of the full-length protein to neuronal actin. This suggests that MT interactions indeed promote Gas2L1 activity towards actin.

The SxAA mutant deficient in EB binding localized similarly to the CH domain, yet it still preferentially stabilized axonal actin (Figs 6I and EV5D; note that a negative ratio in Fig 6I does not indicate a protein's absence from axons, but just reflects the preferential localization to non-axonal neurites). We propose that the SxAA mutation reduces the overall binding of Gas2L1 to MTs, because this binding can occur directly, but also indirectly, through EBs, and the latter is perturbed in the SxAA mutant. Therefore, this mutation reduces the ability of the protein to localize to axons. However, the SxAA-mutated tail fragment of Gas2L1 can still interact with the CH

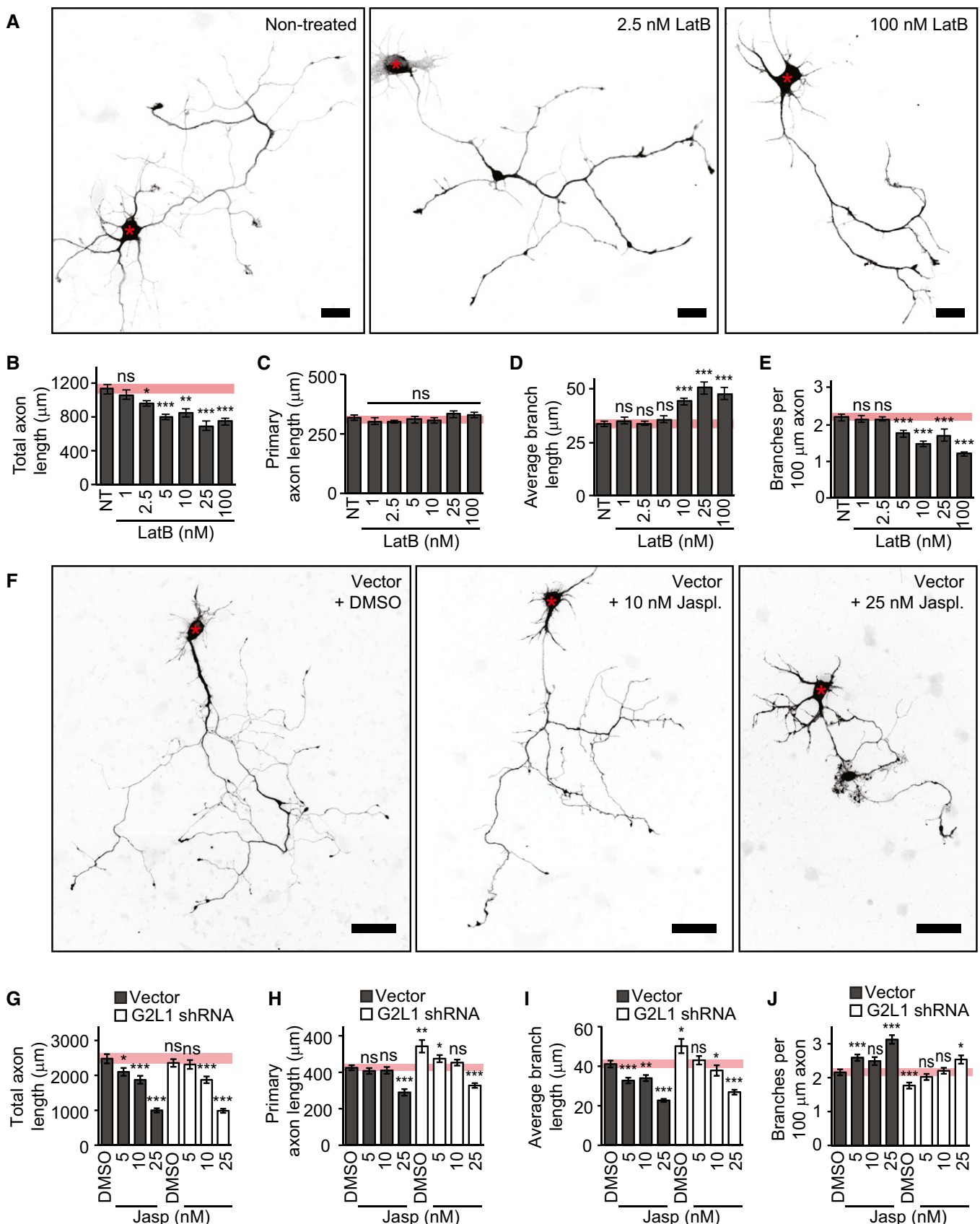

**Figure 5.**

◀

**Figure 5. Gas2L1 phenotypes reflect changes in actin stability.**

A    Silhouettes (from β-galactosidase fill) of DIV3 neurons, transfected at DIV1 and treated with the indicated doses of Latrunculin B for 48 h.
B–E  Quantifications of total axon length (B), primary axon length (C), average axon branch length (D) and the number of branches per 100 μm axon (E) for neurons as described in (A). Pink bars show range of means ± SEM of control conditions. NT = non-treated; n = 37–40 neurons per condition (38/39/37/38/40/39/39 for NT/1/2.5/5/10/25/100 nm LatB, respectively) from three independent experiments.
F    Silhouettes (composite images from β-galactosidase fill) of DIV4 neurons, transfected at DIV1 with empty shRNA vector (grey bars in G–J) or Gas2L1 shRNA (white bars in G–J) and co-expressing β-galactosidase. Twenty-four hours after transfection, neurons were treated with the indicated doses of jasplakinolide or DMSO (equivalent of 25 nM dose) for 48 h. Red asterisks indicate the position of the soma.
G–J  Quantifications of total axon length (G), primary axon length (H), average axon branch length (I) and the number of branches per 100 μm axon (J) for neurons as described in (F). Pink bars show range of means ± SEM of control conditions. n = 64–69 neurons per condition (66/64/68/67 for vector + DMSO/5/10/25 nM jasplakinolide respectively, and 65/66/69/67 for G2L1 shRNA + DMSO/5/10/25 nM jasplakinolide, respectively) from four independent experiments.

Data information: Scale bars: 20 μm in (A), 50 μm in (F). Data are displayed as means ± SEM. Mann–Whitney test, ns: not significant, *P < 0.05, **P < 0.01, ***P < 0.001. For (G–J), all values were compared to those of DMSO control (vector) neurons.

domain and maintains some association with MTs (Figs 3A and EV3), suggesting that the mutated protein can still respond to MT-controlled regulation of autoinhibition, albeit less strongly. As a result, the actin-stabilizing activity of Gas2L1-SxAA is still shifted to the axonal compartment, but not as strongly as that of the wild-type protein.

One could argue that the behaviour of Gas2L1 is simply a result of actin-MT crosslinking via CH and GAR domains, respectively. If this were the case, the MT-binding tail region of Gas2L1 would likely exert a similar effect on other actin-binding CH domains: that is, it would change their localization and actin stabilization patterns. To test this possibility, we fused Gas2L1 Tail to the tandem actin-binding CH domains of α-actinin (ABD and ABD-Tail; Fig EV5C). Although the CH domains of Gas2L1 and α-actinin belong to different classes [34], ABD behaved very similarly to the CH domain of Gas2L1—it preferentially localized to non-axonal neurites, where it also stabilized actin. However, unlike the CH domain of Gas2L1, ABD was not affected by the fusion to Gas2L1 Tail (Fig EV5D–G), indicating that within this fusion, the tail of Gas2L1 makes no direct contribution to the specificity of actin binding. Our data suggest that the MT-binding tail of Gas2L1 can locally and specifically modulate the activity of the CH domain of Gas2L1, likely due to the direct interaction with this domain.

## Discussion

In this study, we show that the cytolinker Gas2L1 regulates axon morphology by stabilizing F-actin. Interestingly, the actin-binding CH domain of Gas2L1 preferentially localizes to non-axonal neurites, but the presence of the MT-binding C-terminus in the full-length protein leads to equal distribution of Gas2L1 to F-actin in all neurites. In spite of its uniform localization, full-length Gas2L1 stabilizes F-actin more potently in the proximal axon, whereas the CH domain preferentially stabilizes F-actin in non-axonal neurites. These results suggest that the binding of Gas2L1 to MTs influences both the localization and activity of the protein in neurons. This finding is further reinforced by the fact that the Gas2L1 MT-binding tail fragment is enriched on MTs in the proximal axon, which coincides with the region of the strongest F-actin stabilization by Gas2L1.

How can MTs regulate the activity of Gas2L1 towards F-actin? We propose that this property depends on the autoinhibition of Gas2L1. Our *in vitro* reconstitution experiments demonstrated that

Gas2L1 efficiently localizes to actin-MT bundles, but not to individual actin filaments or MTs. This can be explained by the interaction between the CH domain and the MT-binding tail, which employs the same interfaces as those used for the actin and MT binding. The association of the Gas2L1 tail domain with MTs is expected to liberate the CH domain and to allow Gas2L1 to interact with actin filaments, placing the activity of the protein under direct control of the simultaneous presence of actin and MTs. Our results complement previous observations hinting towards an inhibitory relation between the CH and GAR domains of Gas2L proteins in other cell types [16,19]. For example, Girdler and colleagues mention observing that Pigs, the *Drosophila* Gas2L ortholog, stabilizes actin at MT interfaces [19]. Therefore, the regulatory mechanism we identified for Gas2L1 may be conserved in evolution. Furthermore, the CH and EF-hand/GAR domains of the *Drosophila* spectraplakin Shot also undergo an autoinhibitory interaction [25]. The direct regulation by actin and MTs might thus be a common mechanism in CH/GAR domain-containing cytolinkers.

Although autoinhibition of Gas2L1 is obvious from the biochemical and *in vitro* reconstitution experiments, it is less apparent in neurons. Purified Gas2L1 does not associate with individual actin filaments *in vitro* unless MTs are present, yet overexpressed Gas2L1 localizes to neuronal F-actin including the areas where MTs are excluded. This discrepancy may arise from the differential affinity of Gas2L1 for various types of actin, e.g. different organization of actin filaments, their post-translational modifications and the presence of actin-associated proteins. Evidence for the binding selectivity of Gas2L1 was already provided by previous studies [18,19]. It is also possible that additional modes of regulation exist in cells, which are not accounted for in the *in vitro* reconstitution experiments. For example, if Gas2L1 autoinhibition is regulated by post-translational modifications such as phosphorylation, the absence of kinases, phosphatases or other regulatory enzymes *in vitro* would account for this discrepancy. Furthermore, in cells, Gas2L1 might interact with additional partners that are not present in our *in vitro* assays and that would skew its activity, for example, by promoting actin binding or by modulating autoinhibitory interactions between the CH domain and the tail region. We note that over time, Gas2L1 does accumulate on actin bundles also in *in vitro* reconstitution experiments, whereas it does not bind MTs, suggesting that the association with actin is the first step towards releasing autoinhibition. Importantly, we find that Gas2L1 localization to neuronal F-actin is not sufficient to stabilize actin. Instead, Gas2L1-mediated actin stabilization appears to depend on the MT binding by its

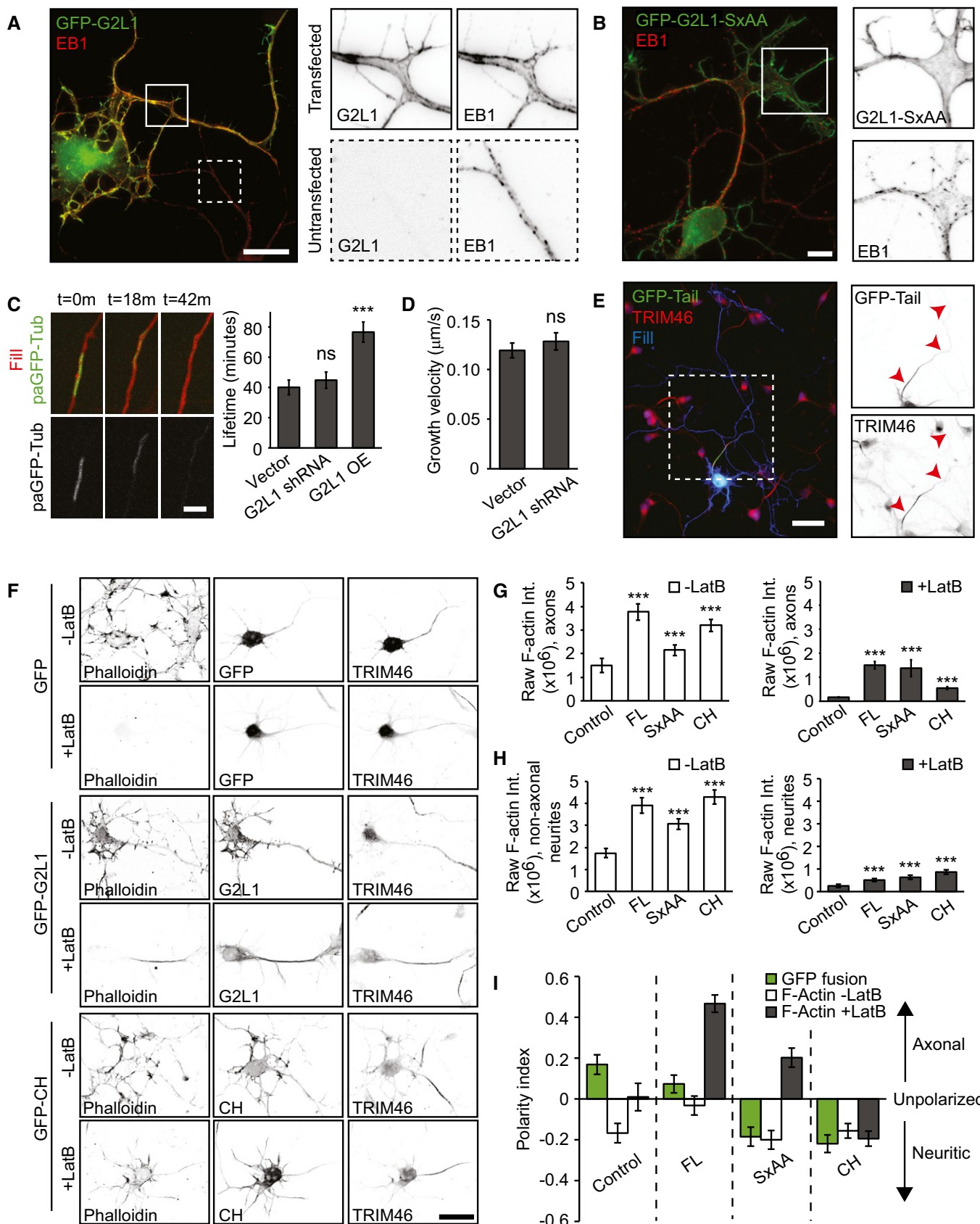

**Figure 6.**

◄

**Figure 6.   MTs modulate the actin-stabilizing activity of Gas2L1.**

A   Endogenous EB1 re-localization upon overexpression of GFP-Gas2L1 (G2L1) in a DIV4 neuron. Boxes indicate zoomed regions of transfected (continuous outline) or untransfected neurons (dotted outline).

B   Endogenous EB1 localization upon overexpression of GFP-Gas2L1-SxAA (G2L1-SxAA) in a DIV3 neuron. Boxes indicate zoomed region of transfected neuron.

C   Lifetime of photoactivated GFP-α-Tubulin signal in axons of DIV3–4 neurons electroporated at DIV0 with RFP, paGFP-αTubulin and control GW1-EV, or Gas2L1 shRNA, or HA-Gas2L1. *n* = 54–59 axons per condition (59 for vector and HA-Gas2L1, 54 for Gas2L1 shRNA) from three independent experiments.

D   Growth velocity of anterograde MT comets near the axon terminus, as labelled with EB3-tagRFP-T in DIV3–4 neurons electroporated at DIV0 with pSuper empty vector (control) or Gas2L1 shRNA. *n* = 27 neurons for control and 28 neurons for Gas2L1 shRNA (664–714 comets) per condition from two independent experiments.

E   Localization of GFP-Tail in DIV3 neurons co-expressing HA-β-galactosidase fill (blue) and stained for TRIM46 (red). Boxed region is enlarged to the right. Arrowheads indicate the axon.

F   DIV3 neurons expressing GFP control, GFP-Gas2L1 or GFP-CH and treated with 10 μM Latrunculin B for 30 min or non-treated, stained for F-actin (phalloidin) and the axon initial segment (TRIM46).

G, H   Raw phalloidin staining intensities in axons (G) and non-axonal neurites (pre-dendrites) (H) of DIV3 neurons expressing the indicated GFP fusions and treated with 10 μM Latrunculin B for 30 min (grey bars) or non-treated (white bars). *n* = 35–40 neurons per condition (40 for GFP, GFP-G2L1, GFP-CH and 35 for GFP-SxAA) from two independent experiments.

I   Polarity index representing localization of the indicated GFP fusions in non-treated neurons (green bars) and localization of F-actin as labelled by phalloidin in DIV3 neurons treated with 10 μM Latrunculin B for 30 min (grey bars) or non-treated (white bars). *n* = 35–40 neurons per condition (40 for GFP, GFP-G2L1, GFP-CH and 35 for GFP-SxAA) from two independent experiments. Positive values indicate axonal enrichment, negative values indicate neuritic enrichment, and a value of 0 signifies no particular enrichment.

Data information: Scale bars: 10 μm in (A, B), 4 μm in (C), 50 μm in (E), 30 μm in (F). Data are displayed as means ± SEM. Mann–Whitney test (C, G, H), unpaired *t*-test (D), ns: not significant, ***$P < 0.001$.

C-terminus, suggesting that full relief of autoinhibition requires the presence of both actin and MTs even in neurons.

We show that the actin-stabilizing drug jasplakinolide is able to rescue axon morphology of Gas2L1-depleted neurons and mimics Gas2L1 overexpression when applied to control neurons. Conversely, actin destabilization via LatB induces morphological changes in line with Gas2L1 depletion: axon branches elongate in response to LatB as they do in Gas2L1-depleted neurons. Interestingly, the primary axon does not respond to low-dose LatB treatment but does become longer after Gas2L1 depletion (summarized in Fig 7A). In line with this observation, axonal branches but not the primary axon are shortened after treatment with low doses of jasplakinolide (5–10 nM). It thus appears that the growth cones of primary axons are less sensitive to mild drug-mediated actin perturbations than axonal branches, possibly due to the presence of more robust regulatory machinery. We speculate that the reduced sensitivity of primary axons to low doses of LatB might be caused by the elevated activity of Gas2L1 in this neuronal compartment, which is caused by a higher affinity of the Gas2L1 C-terminus for MTs in the primary axon. However, we cannot exclude alternative explanations, such as the distinct drug response of specific actin subpopulations that could be present in the primary axon and axonal branches. Precedents for such distinct sensitivity have been described: for example, LatA has been suggested to specifically inhibit the function of the actin regulator β-thymosin and to thereby preferentially inhibit filopodia formation [35].

It is worth emphasizing that the dose of jasplakinolide needed to compensate the effects of neuronal Gas2L1 depletion is minimal (5–10 nM). Together with the dramatic changes induced by Gas2L1 overexpression, these findings suggest that the protein is extremely potent and that endogenous expression levels of Gas2L1 are low. This is in line with other studies investigating Gas2L proteins [16,19,21]. Notably, at early development stages, Pines *et al* [21] reported phenotypes associated with cytoskeletal defects in Pigs null mutant flies without being able to detect the underlying cytoskeletal changes using conventional light microscopy. The nuances of how

Gas2L1 depletion affects the neuronal cytoskeleton are likely similarly subtle and challenging to identify.

Remarkably, key regions of Gas2L1—the actin-binding CH domain, the MT-binding GAR domain and the EB-binding SxIP motif—are also found in the spectraplakin ACF7. Both Gas2L1 and ACF7 are actin-MT crosslinking proteins. However, depletion of ACF7 or its *Drosophila* counterpart Shot impairs axon outgrowth [12,13,36], whereas we find that Gas2L1 depletion promotes axon elongation. Actin-MT crosslinking proteins can thus exert opposite regulatory effects during neuronal development.

The difference between ACF7 and Gas2L1 may be explained by the observation that Gas2L1 is primarily an actin-localized protein, whereas ACF7 strongly interacts with MTs [37,38]. Axon outgrowth requires dynamic MTs to invade the actin-rich growth cone periphery. MTs are disorganized and destabilized in growth cones lacking ACF7 [12,13], and as such ACF7 is thought to guide MT entry into growth cones. We found that MT plus-end growth velocity, MT organization and MT stability appear normal in growth cones of Gas2L1-depleted neurons, suggesting that Gas2L1 does not operate via this mechanism. Instead, we propose that Gas2L1 stabilizes the surrounding actin network in response to the invasion of MTs (Fig 7B).

Although actin and actin-MT coupling are required for axon extension, a subpopulation of actin acts as a barrier for MT invasion in the growth cone and as a brake on outgrowth. The local actin network also produces a contractile, retrograde-oriented force that is counteracted by MTs during extension (reviewed in Ref. [39]). Actin reorganization can thereby positively and negatively affect axon elongation, and the net result is a matter of balance. By increasing actin stability, Gas2L1 could increase the MT-restricting function of growth cone actin and/or the level of contractility. This model explains axon overextension in the absence of Gas2L1, as well as the appearance of buckled MTs in the short axons of Gas2L1-overexpressing neurons (Fig 7B). At endogenous levels, Gas2L1 likely contributes to a certain degree to actin stability required for normal growth cone behaviour. Finally, in contrast to their opposing roles in axon extension, Gas2L1 and ACF7 share a

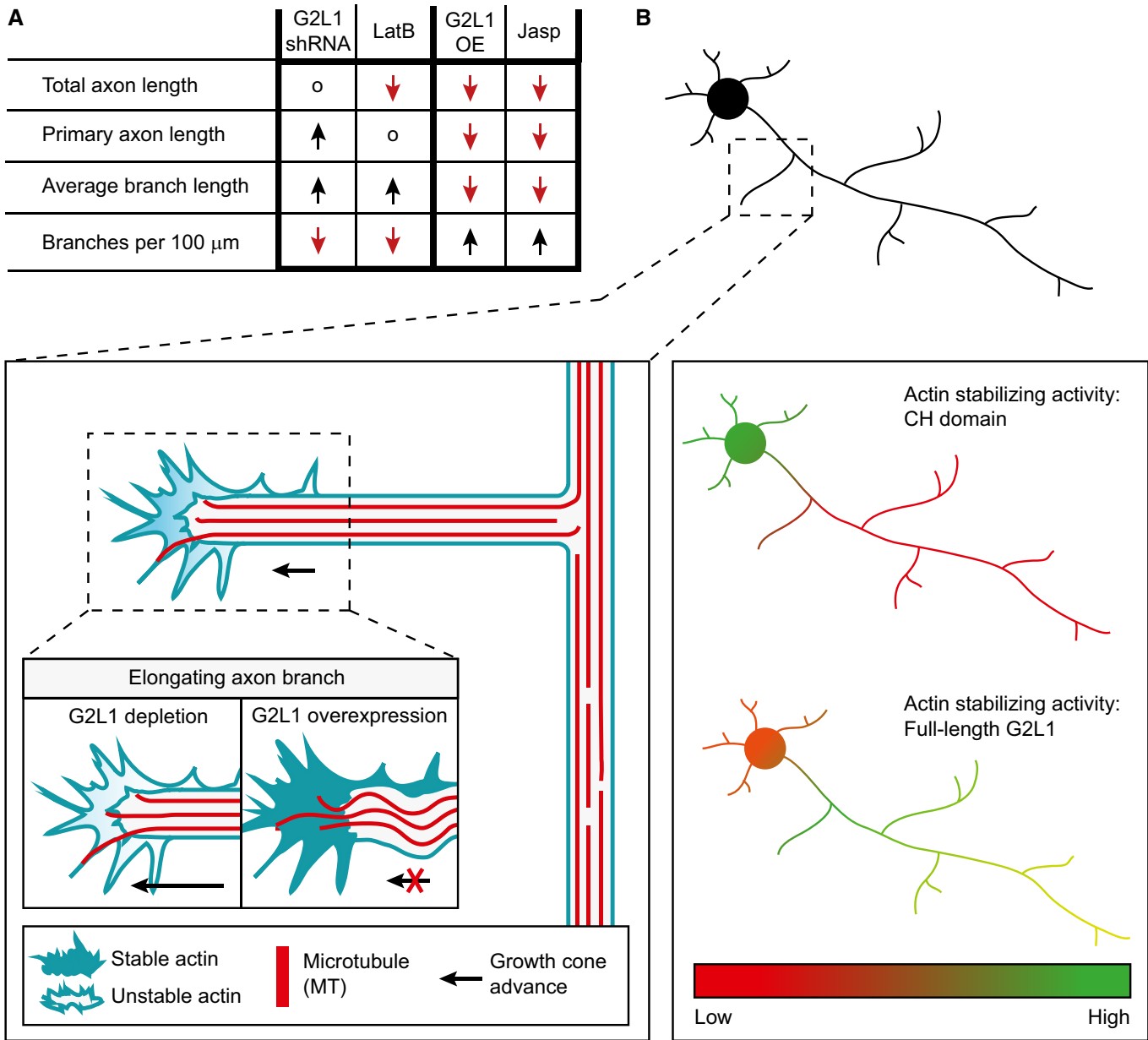

**Figure 7. Model of Gas2L1 activity during axon development.**

A   Summary of the effects of Gas2L1 shRNA-mediated depletion, Gas2L1 overexpression (G2L1 OE) and low nanomolar doses of jasplakinolide (jasp) and Latrunculin B (LatB) on axon development of DIV3–4 neurons. o = no effect; ↓ = lower values, ↑ = higher values.

B   Proposed working model of the role of Gas2L1 during axon branching and outgrowth. MT binding by the tail of Gas2L1 promotes local actin binding and stabilization by the CH domain and shifts the actin-stabilizing activity of Gas2L1 towards the axon. In growth cones, where dynamic MTs probe the actin-rich periphery, Gas2L1 contributes to actin stability levels necessary for normal outgrowth. Reduced actin stability causes overextension of branches in the absence of Gas2L1. Axons of neurons overexpressing Gas2L1 have increased actin stability, resulting in shorter branches with higher F-actin levels and buckling MTs.

function in axon branching [12]. This suggests that both ACF7-mediated MT guidance and actin stabilization via Gas2L1 are necessary for the initial steps of axon branch formation.

To summarize, we reveal the role for the cytolinker Gas2L1 in axon growth and branching, describe important clues about the underlying mechanism and provide insight into the functional diversity of factors involved in cytoskeletal cross-talk.

# Materials and Methods

### Animals, primary neuron cultures, electroporation and transfection

Animal experiments were approved by the Dutch Animal Experiments Committee (DEC) and conducted in agreement with

guidelines of Utrecht University, Dutch law (Wet op de Dierproeven, 1996) and European regulations (Guideline 86/609/EEC). Neuron cultures were derived from hippocampi of E18.5 mixed-sex pups from pregnant Wistar rats (Janvier). Adult rats were at least 10 weeks of age, not involved in previous experiments, provided with unrestricted access to food and water and kept with a companion, wood-chip bedding and paper tissue for cage enrichment. Animals were housed in an environment with a 12-h light–dark cycle and a temperature of $22 \pm 1°C$.

Primary hippocampal dissociated neuron cultures were prepared according to protocols for mechanical/enzymatic dissociation described previously [40]. Neurons were maintained at 37°C and 5% $CO_2$.

In case of electroporation for imaging experiments, 200k neurons per condition were taken from the cell suspension directly after mechanical dissociation, spun down for 5 min at 200 $g$ and resuspended in home-made electroporation buffer (12.5 mM NaCl, 123 mM KCl, 20 mM KOH, 10 mM EGTA, 4.5 mM $MgCl_2$, 20 mM PIPES, pH 7.2). Neurons in electroporation buffer were then mixed with 4 µg DNA per condition, transferred to cuvettes (Bio-Rad GenePulser, 0.2 cm gap) and electroporated using program O-003/"Rat hippocampal neurons" of a Lonza Nucleofector 2b machine. Electroporated neurons were plated in full medium [Neurobasal medium (Gibco) supplemented with 2% B27 (Gibco), 0.5 mM glutamine (Gibco), 15.6 µM glutamate (Sigma-Aldrich) and 1% penicillin–streptomycin (Gibco)] on 18-mm coverslips coated with poly-L-lysine (37.5 µg/ml, Sigma) and laminin (1.25 µg/ml, Roche). The procedure for electroporation for qPCR experiments was described previously [41].

For transfection, 100k neurons per well of a 12-well plate were directly plated in full medium onto 18-mm coverslips coated as described above and transfected at the indicated times using Lipofectamine 2000 (Invitrogen). For one well of a 12-well plate, 1.8 µg DNA was mixed with 3.3 µl Lipofectamine in 200 µl Neurobasal medium and incubated for 30 min at room temperature. Meanwhile, conditioned full medium of neurons was transferred to a new plate, and transfection medium (Neurobasal medium supplemented with 0.5 mM glutamine) was added. The DNA/lipofectamine mix was added to the neurons in transfection medium and incubated for 45 min at 37°C and 5% $CO_2$. After transfection, neurons were rinsed by dipping coverslips in pre-warmed Neurobasal medium and placed back into conditioned full medium.

## Cultured cells and transfection

COS-7 cells were cultured in DMEM/Ham's F10 (1:1) supplemented with 10% FBS and 1% penicillin/streptomycin. Cells were diluted and plated on 18-mm glass coverslips 1 day before transfection. COS7 cells were transfected with DNA constructs using FuGENE6 (Roche) following the manufacturer's protocol.

## DNA and shRNA constructs

The following DNA constructs were used in this study and described before: pβ-actin-HA-β-galactosidase [42], pCI-neo-BirA [43], EB3-tagRFP-T [44] and paGFP-α-Tubulin [45].

All Gas2L1 expression constructs were N-terminal fusions generated from PCR-based strategies using GFP-Gas2L1 or GFP-Gas2L1-SxAA as templates, which were based on human Gas2L1 (GenBank Accession No NM_152236.1) [20]. The following backbones were used to create Gas2L1 plasmids: GW1-GFP/HA for neuronal expression, bioGFP and GW1-HA for pull-down experiments and StrepII-EGFP for protein purification [46]. To create GW1-GFP/HA-ABD and GW1-GFP/HA-ABD-Tail, α-actinin CH domains were amplified from *Gallus gallus* cDNA (amino acids 35–246, NCBI ref. NM_204127.1) using a standard PCR-based strategy. To obtain ABD-Tail, amino acids 35–246 of *Gallus gallus* α-actinin were fused to amino acids 196–681 of human Gas2L1 (NCBI ref. NM_152236.1) by overlap extension PCR. GW2-LifeAct-EGFP was obtained from Dr. M. Adrian and Dr. P. Schätzle (Utrecht University) and based on the LifeAct peptide sequence MGVADLIKKFESISKEE (5′-ATGGG TGTCGCAGATTTGATCAAGAAATTCGAAAGCATCTCAAAGGAAG AA), which was flanked by EGFP inserted into BamHI/XbaI sites of a GW2 backbone.

For shRNA-mediated depletion, shRNA sequences were inserted into HindIII and BglII sites of the pSuper backbone [47]. The following targeting sequences were used: scrambled—GCGCGCTTTGTAGG ATTCG; Gas2L1 shRNA (rat)—CGCCCAATGACATTCGAAA. For use in qPCR experiments, we excised the puromycin resistance region from pSuper.puro and transplanted it into EcoRI/BamHI sites of pSuper-Gas2L1-shRNA.

## Antibodies and reagents

The following antibodies, staining reagents and dilutions were used for immunofluorescence experiments: chicken-anti-β-galactosidase (1:2,500, Aves Labs #BGL-1040); rabbit-anti-TRIM46 serum (1:500, described in Ref. [22]); rabbit-anti-fascin (1:200, Abcam #ab126772); rabbit-anti-cortactin (1:1,000, Abcam #ab81208); rabbit-anti-p34-Arc/ARPC2 (1:250, EMD Millipore #07-227); mouse-anti-alpha-tubulin (1:1,000, Sigma #T-5168, clone #B-5-1-2); mouse-anti-acetylated tubulin (1:500, Sigma #T-7451); mouse-anti-tubulin-tyrosine (1:100, Sigma #T-9028); mouse-anti-EB1 (1:100, BD Biosciences #610535); Alexa Fluor 594 Phalloidin (1:50, Thermo Fischer Scientific #A12381); goat-anti-chicken Alexa 405 (1:400, Abcam # ab175675); goat-anti-chicken Alexa 488 (1:400, Thermo Fischer Scientific #A11039); goat-anti-rabbit Alexa 405 (1:400, Thermo Fischer Scientific #A31556); goat-anti-rabbit Alexa 568 (1:400, Thermo Fischer Scientific #A11036); goat-anti-mouse Alexa 405 (1:400, Thermo Fischer Scientific #A31553), goat-anti-mouse Alexa 568 (1:400, Thermo Fischer Scientific #A11031); goat-anti-mouse-IgG2B Alexa 488 (1:400, Thermo Fischer Scientific #A21141); and goat-anti-mouse-IgG3 Alexa 594 (1:400, Thermo Fischer Scientific #A21155).

The following antibodies, staining reagents and dilutions were used for Western blotting: rabbit-anti-GFP (1:10,000, Abcam #ab290); mouse-anti-HA (1:2,000, BioLegend/Covance #mms-101p, clone #16B12); mouse-anti-actin (1:10,000, Chemicon #MAB1501R, clone #C4); goat-anti-mouse IRDye800CW (1:15.000, LI-COR #926-32210); and goat-anti-rabbit IRDye680LT (1:20,000, LI-COR #926-68021).

The following reagents were used for drug treatments: DMSO (Sigma), Latrunculin B (Santa Cruz Biotechnology #SC-203318) and Jasplakinolide (Tocris/Bio-Techne #2792).

## Immunofluorescence microscopy

For co-stainings with rabbit-anti-fascin, rabbit-anti-p34-Arc/ARPC2 and mouse-anti-EB1, neurons were fixed for 5 min at −20°C in

100% methanol supplemented with 1 mM EGTA, immediately followed by 5 min of fixation at room temperature in 4% paraformaldehyde/4% sucrose. In all other cases, neurons were fixed for 10 min at room temperature in 4% paraformaldehyde/4% sucrose.

After fixation, neurons were washed 2× in PBS. Primary antibodies were diluted in GDB buffer (0.1% BSA, 0.45 M NaCl, 0.3% Triton X-100, 16.7 mM phosphate buffer, pH 7.4) and incubated overnight at 4°C, followed by 3 × 5 min washing in PSB and 1- to 2-h incubation with secondary antibodies in GDB buffer at room temperature. Samples were mounted in VECTASHIELD mounting medium (Vectorlabs).

Images of fixed cells were collected using (i) a Nikon Eclipse 80i upright widefield fluorescence microscope, equipped with a Photometrics CoolSNAP HQ2 CCD camera and Nikon NIS Br software, using one of the following oil objectives: Plan Apo VC 60× N.A. 1.40, Plan Fluor 40× N.A. 1.30 or Plan Fluor 20× N.A. 0.75; or (ii) using a Carl Zeiss LSM 700 confocal laser scanning microscope running ZEN2011 software and using a Plan-Apochromat 63×/1.40 Oil DIC objective. For quantitative comparisons between conditions, imaging settings were kept identical for all acquired images.

## Live cell imaging and photoactivation

Live cell imaging and photoactivation experiments were performed on an inverted Nikon Eclipse Ti-E confocal microscope equipped with a perfect focus system (Nikon), a CSU-X1-A1 Spinning Disc unit (Yokogawa) and a Photometrics Evolve 512 EMCCD camera (Roper Scientific) while using a Plan Apo VC 100× N.A.1.40 oil objective. Neurons were kept at 37°C and 5% $CO_2$ in a stage incubator (Tokai Hit) during imaging. Movies used to determine MT growth velocity were acquired at a frame rate of 1 frame per second, and acquisitions lasted 3 min per movie.

For photoactivation experiments, we made use of an ILas FRAP unit (Roper Scientific) and a Vortran Stradus 405 nm (100 mW) laser. The photoactivation procedure of paGFP-α-Tubulin was previously described [45]. We co-transfected neurons with GW2-mRFP to label transfected neurons and identified axons based on morphology. We placed the soma just outside the field of view, which spanned 34 μm in total, and photoactivated axon regions of approximately 6–7 μm wide in the centre of the image. To prevent excessive photobleaching, frames were acquired at intervals varying between 6 and 15 min.

## qPCR

qPCR experiments were performed with the kind help of M. de Wit and prof. Dr. R.J. Pasterkamp (UMC Utrecht). Experimental procedures are detailed in Ref. [41]. To summarize, neurons were electroporated at DIV0 with empty pSuper-puro or pSuper-Gas2L1 shRNA-puro, and subjected to selection with 0.5 mg/ml puromycin for 48 h prior to RNA isolation. We performed PCRs with the Fast Start DNA Master PLUS SYBR Green I Kit (Roche) and calculated Gas2L1 mRNA levels relative to those of GAPDH and β-actin using the $\Delta\Delta C_t$ method.

The following primer pairs were used: GAPDH 5′-TGC CCCCATGTTTGTGATG and 3′-TGTGGTCATGAGCCCTTCC; β-actin 5′-AGGCCAACCGTGAAAAGATG and 3′-CCAGAGGCATACAGGG ACAAC; Gas2L1 5′-ACATTCGAAACCTGGACGAG and 3′-TCAGCA CCCTCACAAAGATG.

## Pull-down experiments and Western blotting

HEK293 cells were cultured in 50/50 DMEM/Ham's F10 medium (DMEM: Lonza or Biowest; Ham's F10: Lonza) supplemented with 10% FCS (Sigma) and 1% penicillin–streptomycin (Sigma), and routinely tested for mycoplasma.

HEK293 cells were co-transfected with pCI-Neo-BirA, bioGFP-tagged constructs and HA-tagged constructs using MaxPEI (Polysciences) in a ratio of 3:1 PEI:DNA. After 24- to 48-h expression, cells were washed with ice-cold PBS supplemented with 0.5× protease inhibitor cocktail (Roche) and 5 mM $MgCl_2$. Next, cells were lysed in pull-down lysis buffer for 30 min on ice [50 mM Tris–HCl pH 7.5, 233 mM NaCl, 0.5% Triton X-100, 5 mM $MgCl_2$, protease inhibitors (Roche)]. Lysates were then pre-cleared by centrifugation (15 min at 13.2 krpm, 4°C), and supernatants were incubated with pre-blocked magnetic bioGFP beads (Invitrogen Dynabeads M-280 Streptavidin; blocking by incubation with 20 mM Tris–HCl pH 7.5, 150 mM KCl, 0.2 μg/μl chicken egg albumin for 1 h at 4°C) for 1.5–2 h at 4°C. Lastly, beads were generously washed 5× in wash buffer (20 mM Tris–HCl pH 7.5, 150 mM KCl, 0.1% Triton X-100) and eluted by boiling 10 min at 95°C in 2× DTT+ sample buffer (20% glycerol, 4% SDS, 200 mM DTT, 100 mM Tris–HCl pH 6.8, bromophenol blue).

For pull-down experiments in the presence of Latrunculin B, 10 μM Latrunculin B was added to HEK293 cells 30 min prior to lysis. Lysis buffers used during the experiment were additionally supplemented with 10 μM Latrunculin B.

Samples ran on 12–14% SDS–PAGE gels and were transferred onto nitrocellulose membranes by semi-dry blotting at 16 V for 1 h. Membranes were blocked in 2% bovine serum albumin (BSA)/ 0.02% Tween 20/PBS for 1 h at room temperature, followed by overnight incubation with primary antibodies in 2%BSA/0.02% Tween20/PBS, 3 × 5 min washes with 0.02% Tween20/PBS and 1 h of incubation with secondary antibody in 2%BSA/0.02% Tween20/PBS, followed by another three washing steps.

Membranes were scanned on an Odyssey Infrared Imaging system (LI-COR Biosciences) and, if required, re-incubated to detect and discriminate multiple signals on the same membrane.

## Gas2L1 protein purification

Gas2L1 fusions were purified from HEK293 cells using Strep(II)-Strep-Tactin affinity purification. We expressed tandem StrepII-tagged GFP-Gas2L1 fusions for 24–48 h in HEK293 cells cultured and transfected as described for pull-down experiments.

Prior to lysis, cells were washed once in ice-cold PBS supplemented with 0.5× protease inhibitors. Cells were lysed for 15 min on ice in purification lysis buffer (50 mM HEPES pH 7.4, 300 mM NaCl, 0.5% Triton X-100, 1× protease inhibitors), followed by centrifugation for 20 min at 13.2 krpm/4°C. Supernatants were incubated for 45 min in the presence of pre-washed Strep-Tactin Sepharose High Performance beads (GE Healthcare) at 4°C. Following incubation, beads were washed three times in ice-cold wash buffer (50 mM HEPES pH 7.4, 300 mM NaCl, 0.5% Triton X-100). StrepII-GFP-Gas2L1 was then eluted in standard elution buffer

(50 mM HEPES pH 7.4, 150 mM NaCl, 1 mM MgCl$_2$, 1 mM EGTA, 0.05% Triton X-100, 1 mM DTT, 2.5 mM d-Desthiobiotin) by incubating beads with elution buffer for 10 min on ice. To prevent aggregation, the elution buffer used to purify Gas2L1 truncation mutants was supplemented with 50 mM arginine, 50 mM glutamic acid and 10% glycerol. After incubation with elution buffers, supernatants were collected by centrifuging for 1 min at 800 $g$/4°C and cleared a second time by centrifuging for 1 min at 13.2 krpm/4°C. The remaining supernatant was aliquoted in single-use vials, snap-frozen in liquid nitrogen and stored at −80°C.

The concentration and purity of purified Gas2L1 fusions was determined by Coomassie blue staining of SDS–PAGE gels. Set amounts of purified Gas2L1 ran alongside a range of standardized BSA concentrations (100–2,000 ng). The resulting Gas2L1 band intensities were compared to the BSA concentration curve and used to calculate the amount of protein in ng/µl from which molarity could be determined.

### In vitro reconstitution assays

#### Protein production and purification

Gas2L1 was purified as described above. Lyophilized porcine brain tubulins were obtained from Cytoskeleton (Denver, CO, USA), resuspended at 50–100 µM in MRB80, snap-frozen and stored at −80°C until use. G-actin was purified from rabbit skeletal muscle acetone powder [48,49] and stored at −80°C in G-buffer [23]. Before use, G-actin was thawed overnight at 4°C and spun for 15 min at 149,000 × $g$ to remove any aggregates, and stored at 4°C for no longer than 2 weeks. Alexa Fluor 647 (Molecular Probes, Life Technologies, Carlsbad, CA, USA) was used to produce labelled G-actin [48]. 6xHis-tagged recombinant human mCherry-EB3 was expressed and purified as described before [50–52].

#### Preparation of flow cells for in vitro assays

Glass coverslips and slides (Menzel-Gläser, Braunschweig, Germany) were cleaned in base-piranha and stored at RT in Milli-Q for no longer than 5 days. Flow cell channels (10–15 µl) were assembled by cutting parafilm in thin pieces, which were sandwiched between clean glass slides and coverslips [23]. Biotinylated glass surfaces were obtained by sequentially incubating the flow cell channels with the following solutions: 0.1 mg/ml PLL-PEG-Biotin (PLL(20)-g[3.5]-PEG(2)/PEG(3.4)-Biotin(20%), SuSos AG, Dübendorf, Switzerland) for 30–60 min, 0.25 mg/ml streptavidin (Thermo Scientific Pierce Protein Biology Products, Rockford, IL, USA) for 10 min, 0.5 mg/ml κ-casein for 10 min and 1% ($w/v$) Pluronic F-127 for 10 min, all diluted in MRB80, with 40–70 µl rinses with MRB80 in between steps.

#### TIRF assays

Stabilized microtubule seeds were prepared using the slowly hydrolysable GTP analogue guanylyl-(α,β)-methylene-diphosphonate (GMPCPP, Jena Biosciences, Jena, Germany), following the double-cycling as described in Ref. [23] and using 12% labelled tubulin, 18% biotinylated tubulin and 70% unlabelled tubulin. Phalloidin-stabilized F-actin was first polymerized at 7.5 µM G-actin concentration (15% labelled and 85% unlabelled G-actin) in MRB80 (with 50 mM KCl, 0.2 mM Mg-ATP, 4 mM DTT) for 30–90 min at room temperature, before adding phalloidin in 1:1 molar ratio. Dynamic

microtubules were nucleated from GMPCPP-stabilized microtubule seeds bound to the biotinylated surface of the glass flow cells. Any non-attached seeds were always rinsed with MRB80 before the actin filaments and/or the microtubule polymerization mix was added. Once the flow cell channel was prepared with microtubule seeds, the microtubule polymerization mix, including actin filaments, was added. The core reaction mix is MRB80 buffer-based (pH 6.8 with KOH, 80 mM PIPES, 4 mM MgCl$_2$, 1 mM EGTA) and supplemented with 0.5 mg/ml κ-casein, 0.2% (v/v) methyl cellulose, 75 mM KCl, 1 mM GTP, 0.2 mM Mg-ATP and an oxygen scavenging system [4 mM dithiothreitol (DTT), 2 mM protocatechuic acid (PCA) and 100 nM protocatechuate-3,4-dioxygenase (PCD)].

The tubulin concentration was kept at 20 µM (6% labelled and 94% unlabelled tubulin), and the pre-polymerized F-actin (15% labelled and 85% unlabelled G-actin) was diluted to a final concentration of 10 nM–1 µM. After mixing (before the addition of the F-actin), the final mixture was clarified at 149,000 × $g$ for 5 min and immediately added to the flow cell channel, which was finally sealed with vacuum grease to avoid solvent evaporation while imaging.

#### TIRF microscope (laser, power, time lapses, temperature)

Triple-colour TIRF microscopy experiments were performed on a Nikon Eclipse Ti-E inverted microscope (Nikon Corporation, Tokyo, Japan) equipped with an Apo TIRF 100 × 1.49 N.A. oil objective, a motorized stage, Perfect Focus System, a motorized TIRF illuminator (Roper Scientific, Tucson, AZ, USA) and a QuantEM:512SC EMCCD camera (Photometrics, Roper Scientific). For excitation, we used a 561 nm (50 mW) Jive (Cobolt, Solna, Sweden), a 488 nm (40 mW) Calypso (Cobolt) diode-pumped solid-state laser and a 635 nm (28 mW) Melles Griot laser (CVI Laser Optics & Melles Griot, Didam, Netherlands). Most of the imaging of dynamic microtubules was performed at 3 s per frame with 50–100 ms exposure time at 8–18% laser power. The sample temperature was kept fixed with the use of a home-built objective heater/cooler to 30 ± 1°C.

### Data analysis and image processing for biochemical assays and cell imaging experiments

All images were processed with ImageJ software and plugins described below. For display purposes, images shown in this publication had their dynamic ranges adjusted when necessary. Quantitative analyses were performed on raw data collected using the same microscope with identical settings throughout experiments.

For axon morphology analyses, β-galactosidase co-overexpression and staining was used to identify transfected neurons and to visualize morphology. We acquired composite images of the total axon by manually piecing together multiple acquisitions of the same neuron. TRIM46 staining was included to identify axons, which were traced and analysed using the NeuronJ plugin for ImageJ by Dr. E. Meijering [53]. We excluded neurons with multiple axons and neurons without TRIM46-positive neurites from our analyses. To avoid selection bias, neurons that satisfied these requirements were included on a first-come-first-served basis during image acquisition. The number of independent experiments stated in figure legends reflects the number of experiments performed on independent neuron cultures, i.e. cultures derived from pups from a different mother animal. We would like to note that, as is expected of

primary cultures, there was some morphological heterogeneity between individual neurons and also between neuron cultures.

To quantify the growth cone area and the ratio of acetylated/tyrosinated tubulin, neurons were co-transfected with β-galactosidase and stained for β-galactosidase (blue), acetylated tubulin (green) and tyrosinated tubulin (red). Transfected neurons were identified from β-galactosidase staining, and images were acquired in three channels with identical microscope settings for all coverslips within one experiment. Using ImageJ, growth cones of transfected neurons were drawn and their area was measured. Next, the intensity in the green (acetylated tubulin) and red (tyrosinated tubulin) channels was measured for all growth cones of transfected neurons that did not cross untransfected neurons. The raw intensities were used to calculate the ratio acetylated tubulin/tyrosinated tubulin.

For quantification of MT orientation inside growth cones, neurons were co-transfected with β-galactosidase and stained for α-tubulin. MT orientation in growth cones of transfected neurons was analysed from α-tubulin staining and classified into straight, buckled or fan-like phenotype.

To quantify MT growth velocity in growth cones, we co-transfected neurons with shRNA, EB3-tagRFP-T and a GFP fill. Axons were identified on the basis of morphology, and growth cones could be located by observing the GFP signal. Kymographs were drawn along the shafts and growth cones, using a line thickness sufficient to cover the majority of EB traces as seen on a maximum intensity projection, and generated using the KymoResliceWide plugin for ImageJ by Dr. E. Katrukha (Utrecht University, available on GitHub). The resulting kymographs were manually traced, and growth velocities obtained from individual traces were first used to calculate average velocity per neuron/kymograph before using the resulting values to calculate the average growth velocity in each condition.

For polarity index calculations, neurons were transfected with different GFP-Gas2L1 fusion constructs. TRIM46 staining was used to identify the axon initial segment, and Phalloidin staining was used to visualize F-actin. Images were acquired with identical microscope settings for all coverslips within one experiment. Using ImageJ, a line of 3.18 μm width with a length of 12–32 μm was drawn in the axon initial segment. The same ROI was positioned in the nearest area where no cell was present to obtain background intensity, which was subtracted from the raw measurements. Intensities were measured at identical positions in both the green (GFP-Gas2L1 fusions) and red channels (Phalloidin). The same was done for two dendrites of the same neuron. Polarity indexes were then calculated using the formula = $(Ia - Id)/(Ia + Id)$, in which $Ia$ is the intensity in the axon and $Id$ is the average intensity of two dendrites.

**Data analysis for *in vitro* assays**

For the *in vitro* reconstitution assays, all image processing was performed in FIJI, including the construction of kymographs [54]. Kymographs were constructed using the plugin in ImageJ. For the intensity ratios, we used a custom-written program in Python. Briefly, we selected a region of interest and measured the intensity distributions of both the GFP signal and the actin signal. The intensity ratios were calculated using the formula = $(I_{GFP} - I_{camera\ noise})/\langle I_{actin}\rangle$,

where the $I_{camera\ noise}$ is the peak GFP-intensity measured in an empty channel, and $\langle I_{actin}\rangle$ is the mean actin intensity of the selected region.

**Statistics**

Statistical analysis was performed in GraphPad 5. Datasets were first tested for normality using Shapiro–Wilk tests. Datasets that did not satisfy the normality assumption were compared using Mann–Whitney *U*-test, and datasets that did were compared using unpaired *t*-tests. Tests were two-tailed. Statistical tests used for each experiment are detailed in the figure legends.

**Expanded View** for this article is available online.

## Acknowledgements

qPCR experiments were performed with the kind help of M. de Wit and support of Prof. Dr. R.J. Pasterkamp. We thank M. Vinkenoog-Kuit for actin purification. This work was supported by the European Research Council (Synergy grant 609822 to M.D. and A.A. and ERC Consolidator Grant 617050 to C.C.H.), the Netherlands Organization for Scientific Research (NWO-ALW-VICI 865.10.010 to C.C.H.), the Netherlands Organization for Health Research and Development (ZonMW-TOP 912.16.058 to C.C.H.) and the Hong Kong Research Grants Council (General Research Fund and Theme-based Research Scheme to R.Z.Q.).

## Author contributions

DW and JJAH performed, designed and analysed biochemical and imaging experiments in cells. CA performed, designed and analysed *in vitro* reconstitution experiments. DW and OIK performed qPCR experiments collaboratively. FKCA and RZQ provided reagents and feedback. MD, GHK, CCH and AA supervised the research and coordinated the study. DW, CA and AA wrote the article.

## Conflict of interest

The authors declare that they have no conflict of interest.

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
