## [Review Process File · EMBO Reports]

Cytolinker Gas2L1 regulates axon morphology through microtubule-modulated actin stabilization

Dieudonné van de Willige, Jessica J.A. Hummel, Celine Alkemade, Olga I. Kahn, Franco K.C. Au, Robert Z. Qi, Marileen Dogterom, Gijsje H. Koenderink, Casper C. Hoogenraad, Anna Akhmanova

Review timeline:

Submission date:	16 January 2019
Editorial Decision:	19 February 2019
Revision received:	13 July 2019
Editorial Decision:	5 August 2019
Revision received:	7 August 2019
Accepted:	15 August 2019

Editor: Deniz Senyilmaz-Tiebe

Transaction Report:

1st Editorial Decision

19 February 2019

Thank you for submitting your manuscript for consideration by EMBO Reports. It has now been seen by three referees whose comments are shown below.

As you can see, all referees express interest in the proposed function of Gas2L1 in regulating axon morphology. However, they also raise concerns that need to be addressed in full before we can consider publication of the manuscript here.

Given these constructive comments, I would like to invite you to revise your manuscript with the understanding that the referee must be fully addressed and their suggestions taken on board. Please address all referee concerns in a complete point-by-point response. Acceptance of the manuscript will depend on a positive outcome of a second round of review. It is EMBO Reports policy to allow a single round of revision only and acceptance or rejection of the manuscript will therefore depend on the completeness of your responses included in the next, final version of the manuscript.

REFeree REPORTS

Referee #1:

Interesting and important findings of Gas2L1 in neurons, indicating it stabilizes actin filaments in a way that's auto-uninhibited by binding to microtubules, with functional relevance to the preference

for axon branching over growth. The drug parts are a bit weak, with the actin drug somewhat phenocopying the Gas2L1 knockdown but not quite really, and the lack of any nocodazole (or other anti-microtubule drug) experiment to prove the point that without microtubules the Gas1L1 wouldn't stabilize actin. I realize that it's hard to completely rid the neurons of microtubules but long treatments with nocodazole or vinblastine could clear out most of the microtubules. None of the drug stuff is really necessary, if the results are equivocal. Molecular work, in vitro work and imaging work are well done.

Referee #2:

Manuscript Number: EMBOR-2019-47732-T

"Cytolinker Gas2L1 regulates axon morphology through microtubule-modulated actin stabilization."
Van de Willige et al.

This is an interesting manuscript that characterizes the potential function in neurons of the cytolinker protein Gas2L1, thought to specifically stabilize axonal actin filaments somehow mediated by the interaction with microtubules. The paper appears quite advanced and contains a significant amount of data, most of them quite solid, in particular the in vitro work and biochemical experiments. The autoinhibition model presented by the authors is quite appealing, although it appears to mostly work in vitro, and its relevance in vivo is less clear, as mentioned by the authors, which could be considered as one weakness in the line of argument concerning the precise function of this protein in neurons. This brings me to the cell biology, which is certainly less clear in places than the rest of the work, in particular when it comes to functional interference or over-activation of Gas2L1 and the precise outcomes of such treatments. I admit that this is mostly due perhaps to the difficulty with working with primary neurons, which are also not entirely appealing concerning work on the actin cytoskeleton (difficult to handle in this respect), but I will provide some specific suggestions for improvements on the cell biology part of the paper for the authors to consider (see below), in order to make the overall conclusions of the paper more comprehensible for the average reader.

Specific Critique:

1) The knockdown experiments look interesting at first glance, and quite a few experiments in the paper rely on the assumption that shRNA-mediated downregulation of G2L1 mRNA (as assessed by qPCR) does indeed also correlate with a significant reduction of G2L1 at the protein level. This is not shown anywhere and I think the authors should try to confirm this by preparing enough material for Western Blotting. Showing a reduction at the protein level would make conclusions on specific effects of the knockdown much stronger, in particular those that include the lack of effects on, as for instance on the velocity of MT lifetime (Fig. 6B) and MT growth (Fig. 6C).

2) In the first part of the results, the authors also mention that Gas2L1 overexpression induces excessive filopodia and lamellipodia-like structures (which is supposed to be shown in Fig. S1C), but the displayed images are not convincing enough in this respect to justify such a statement. The authors should use marker proteins for the mentioned structures (such as fascin for filopodia shafts or VASP for filopodia tips as well as cortactin and Arp2/3-complex for lamellipodia) in these cultures, which would make such a statement much more reliable.

3) Figure 2 looks very clear, but I was confused by a mixup I think in the images displayed in Supplementary Fig. S2 as compared to both text and legends, since S2E is labeled as control experiment showing that in the absence of Gas2L1, there is no co-alignment of MT and actin bundles. However, the image in S2E clearly shows the presence of bundles and G2L1, but also EB3, so I think the authors have erroneously swapped the images shown in S2E with what's coming next in S2F-H. This must be corrected!

4) In Figs 5 and 6, the authors quite systematically analyze combined effects of treatments with different concentrations of LatB or jasplakinolide and G2L1 knockdown or overexpression etc, which is interesting and informative, of course, to certain extent, but also quite indirect! I guess I would have expected a few more of the type of experiments shown in Figure 4, asking

much simpler questions, as for instance: How does overexpression or knockdown of G2L1 affect actin dynamics in the axon and in growth cones? In Fig. 4C, the authors show an interesting buckling of MT in growth cones, and perhaps increased growth cone area using lifeact expression. Can the authors expand on this? For instance, show some informative video microscopy data, and provide some quantification on this? Finally, can the authors explore proposed differential F-actin stabilities in different conditions of GsL1 manipulation by performing a photoactivation activation experiment of actin in different subcellular regions (as for instance in the axon), in analogy to what they have done with tubulin in Fig. 6B? Such experiments would be much more direct than the combinations of inhibitor treatments mentioned above, which could then perhaps be shifted to the Supplement and be replaced by more direct cell biological expts as suggested.

One more point: I can't really discern the claimed displacement of EB1 upon overexpression of G2L1 in Figure 6A, or the mentioned re-targeting of EB1 to actin. This has to be demonstrated more clearly by counter-stainings with MTs and the actin system, using individual channels for each.

5) On the positive side, the control experiment with using the actin-binding domain of alpha-actinin to replace the CH-domain in G2L1 is quite interesting, but in the end, I don't quite grasp what the authors mean with: "the localization of Gas2L1 thus appears to be an emergent property of the combination of its specific CH-domain and MT-binding tail" (last sentence of the Results section)? This is too vague, hence does not mean a lot! Is it really the specific interaction that's relevant or is there anything else? The problem is that the MT-binding C-terminus used is very large, so how can the authors exclude additional interactions with other proteins to play relevant functions? They should at least mention this! If the authors want to make sure that MT binding alone is what's important, they should narrow down the MT interaction surface or screen for point mutants to interfere with this much more specifically, which would then help to substantiate such a statement! It might also be worth constructing something like a "mini-G2L1" solely harboring the actin and MT binding surfaces, and then explore the effects of that one, which would demonstrate whether or not additional interactions are really relevant for the specific functions of G2L1 proposed.

Minor comments:

6) I couldn't find Supplementary Figures S3 and S4 until I realized that they were probably not there because each Supplementary Figure fitted the main Figure named accordingly, but this seemed quite strange to me...

7) On page 14, third para, the text should read ...Gas2L1 likely contributes to a certain level to (not "of") actin stability required for normal growth cone behavior.

Referee #3:

Willige et al explore the role and mechanism of action of the actin-microtubule linker Gas2L1 in neurons. The authors combine in vitro reconstitution assays, structure function studies and pull-downs with gain and loss of function studies and pharmacology in primary neuronal culture to further pinpoint the mechanism of action. The authors demonstrate that Gas2L1 promotes the formation of branches along the axon while restricting elongation of the axon in developing hippocampal neurons. They propose that these growth regulatory roles of Gas2L1 occur by stabilising acting a function that requires interaction with the microtubule cytoskeleton and overcoming an autoinhibitory mechanism.

Overall, I like the data and the combination of in vivo with in vitro reconstitution approaches. I find that the authors provide satisfactory data to support most of their arguments and their findings will be of interest to a broad range of reader interested in cytoskeleton regulation in neurons and belong. The paper could improve from few corrections, further discussion and perhaps a couple of extra analysis.

General comment:

- A key conclusion of the authors is that Gas2L1 regulates neuronal morphology by stabilising the actin cytoskeleton. This conclusion is based on localisation studies, similarity of phenotypes when compared to the effects of actin destabilisation drugs and the rescue with jasplakinolide. However,

several microtubule actin linkers have been reported necessary for MT bundle formation, in their absence, MTs acquired unbundle and misdirected trajectories and fail to extend into growth cone filopodia. To rule out the contribution of Gas2L1 to the regulation of MTs, the authors test the MT stability and the speed of polymerization but do not comment on MT organisation particularly at the growth cone. A set of analysis aiming to describe the bundle organisation of MTs would be very helpful in order to rule out MT regulation.

- A further conclusion of the authors is that the interaction of Gas2L1 with MTs is able to direct its actin stabilization function. The authors discuss that this property depends on the autoinhibition of Gas2L1 and propose in the discussion that the association of the Gas2L1 tail domain with MTs is expected to liberate the CH domain and to allow Gas2L1 to interact with actin filaments. This comes as a surprise since there are previous discussions suggesting the opposite, as an example please see Pg. 8 "The localization of Gas2L1 to F-actin was also apparent in subcellular areas devoid of MTs. This observation reveals that in contrast to our in vitro experiments, Gas2L1 can localize to actin structures independently of MTs in neurons. This result further strengthens the idea that actin binding may be the first step towards stably relieving the autoinhibition of Gas2L1". This contradiction could require further clarification.

Specific comments:

- Pg. 6 "By contrast, the Tail mutant was able to bind MTs in the absence of actin filaments (Fig. 2F)" --- Tail mutant or Tail domain?

- Pg.6 "In a composite assay where more actin filaments were available for binding, we observed MTs covered with multiple co-aligned actin filaments (Fig. 2H), whereas no MT-actin co-alignment occurred without Gas2L1 (Fig. S2E)" --- S2E is in the presence of G2L1, do the authors refer to S2E?

- Pg.8 "...the addition of 10 μ M Latrunculin B (LatB), which inhibits actin polymerization, abolished the interaction between actin and the CH domain (Fig. 3D). This reveals that our pull-down experiments reflect an interaction between Gas2L1 and actin filaments rather than actin monomers" --- Could the authors comment on the finding that full length Gas2L1+ CH domain ability to pull down actin when coexpressed in not abolished with 10 μ M Latrunculin B?

- Fig.S2 Potential levelling mistake: F-H TIRF images (F) (or H??) and kymographs (G, H) (or F, G) showing specific Gas2L1 localization to MT-actin overlaps (E instead of H?) and absence of plus-end tracking in an in vitro reconstitution with Gas2L1, F-actin (1 μ M), MTs and EB3 (is the first panel in F tubulin? If so mark it in accordance). These data indicate that EB3 does not influence the localization of Gas2L1 in this system, even when EB3 tracks growing MT plus ends (H) (or G?).

- Pg. 9 "... These results are in agreement with our observation that Gas2L1 is not a plus-end tracking protein in vitro. However, Gas2L1 does not recruit EB3 in vitro (Fig. S2H)...." --- S2H or S2G?

- Pg.11 when describing the role of MTs in Gas2L1 activity, the authors compare GFP-Gas2L1-SxAA with GFP-CH. GFP-CH tends to localise to dendrites the position where it protects actin. However GFP-Gas2L1-SxAA also equally localises to dendrites but surprisingly fails to protect dendritic actin and instead protects axonal actin. The authors comment by saying that "... SxAA mutant displayed slightly less efficient axonal localization and actin stabilization (Fig. 6H), implying that the SxIP motif of Gas2L1 may function to enhance the affinity of the protein for axonal MTs" but the stamen does not represent the data presented and instead fails to support the hypothesis. This would need further clarification.

1st Revision - authors' response

13 July 2019

REVIEWER COMMENTS

Referee #1:

Interesting and important findings of Gas2L1 in neurons, indicating it stabilizes actin filaments in a way that's auto-uninhibited by binding to microtubules, with functional relevance to the preference for axon branching over growth. The drug parts are a bit weak, with the actin drug somewhat phenocopying the Gas2L1 knockdown but not quite really, and the lack of any nocodazole (or other anti-microtubule drug) experiment to prove the point that without microtubules the Gas1L1 wouldn't stabilize actin. I realize that it's hard to completely rid the neurons of microtubules but long treatments with nocodazole or vinblastine could clear out most of the microtubules. None of the drug stuff is really necessary, if the results are equivocal. Molecular work, in vitro work and imaging work are well done.

We thank the reviewer for the positive assessment of our work. Unfortunately, it is not possible to test the effect of complete microtubule removal on the actin-stabilizing activity of Gas2L1, because microtubule depolymerisation is well-known to have a profound effect on the actin cytoskeleton, for example, by affecting signalling by Rho GTPases (e.g. by activating the microtubule-bound RhoGEF, GEF-H1 [1]). Indeed, 2 hr treatment with 10 μ M nocodazole had a strong effect on the cell morphology and actin organization, (Figure for Reviewers 1). Microtubule depolymerisation appears to result in an increase in F-actin, as was observed from staining with phalloidin. This was especially clear in the cell soma of nocodazole-treated cells. In neurites of control neurons, F-actin is mainly seen forming actin patches, whereas in nocodazole-treated neurons F-actin forms longer stretches.

Figure for Reviewers 1

Representative images of DIV3 neurons treated with 10 μ M nocodazole for 2 hours and non-treated control neurons, stained for MTs (α -tubulin) and F-actin (Phalloidin). Several cell somas are indicated by red arrows. Panels on the right are F-actin zooms of the boxes, showing actin patches and stretches for non-treated control and nocodazole treated neurons, respectively.

Data information:

Scale bar: 30 μ m

Referee #2:

This is an interesting manuscript that characterizes the potential function in neurons of the cytolinker protein Gas2L1, thought to specifically stabilize axonal actin filaments somehow mediated by the interaction with microtubules. The paper appears quite advanced and contains a significant amount of data, most of them quite solid, in particular the in vitro work and biochemical experiments. The autoinhibition model presented by the authors is quite appealing, although it appears to mostly work in vitro, and its relevance in vivo is less clear, as mentioned by the authors, which could be considered as one weakness in the line of argument concerning the precise function of this protein in neurons. This brings me to the cell biology, which is certainly less clear in places than the rest of the work, in particular when it comes to functional interference or over-activation of Gas2L1 and the precise outcomes of such treatments. I admit that this is mostly due perhaps to the difficulty with working with primary neurons, which are also not entirely appealing concerning work on the actin cytoskeleton (difficult to handle in this respect), but I will provide some specific suggestions for improvements on the cell biology part of the paper for the authors to consider (see below), in order to make the overall conclusions of the paper more comprehensible for the average reader.

We thank the reviewer for the helpful suggestions.

Specific Critique:

1) The knockdown experiments look interesting at first glance, and quite a few experiments in the paper rely on the assumption that shRNA-mediated downregulation of G2L1 mRNA (as assessed by qPCR) does indeed also correlate with a significant reduction of G2L1 at the protein level. This is not shown anywhere and I think the authors should try to confirm this by preparing enough material for Western Blotting. Showing a reduction at the protein level would make conclusions on specific effects of the knockdown much stronger, in particular those that include the lack of effects on, as for instance on the velocity of MT lifetime (Fig. 6B) and MT growth (Fig. 6C).

We used different amounts of rat cortical neuron extracts, ran these alongside a lysate of HEK293 cells expressing HA-Gas2L1 and tried to detect the endogenous Gas2L1 band using two different antibodies (a commercial antibody and a custom-made antibody described previously [2]). Unfortunately, despite our best efforts, we could not detect a specific band corresponding to the endogenous Gas2L1 in neuronal extracts (Figure for Reviewers 2), which is consistent with the published literature, where Gas2L1 protein was reported to be expressed only at low levels [3-5]. We thus have to rely on qPCR to show that Gas2L1 is expressed and depleted in our neurons. Importantly, we do rescue the phenotypes observed after depletion of Gas2L1 by expressing tagged Gas2L1 constructs, and we think that this provides support for the validity of our conclusions.

Figure for Reviewers 2

Western blot showing lysates derived from different amounts of cortical neurons (lane 1: 1.2 million neurons, lane 2: 2.4 million neurons, lane 3: 3.6 million neurons) and a lysate of HEK293 cells expressing HA-G2L1 (lane 4), probed with antibodies against G2L1 (left panel: antibody as

described in Au et al., 2017 [2], right panel: Atlas antibodies #HPA019858) and actin (bottom panels). Red arrowheads indicate the band corresponding to full-length Gas2L1.

2) In the first part of the results, the authors also mention that Gas2L1 overexpression induces excessive filopodia and lamellipodia-like structures (which is supposed to be shown in Fig. S1C), but the displayed images are not convincing enough in this respect to justify such a statement. The authors should use marker proteins for the mentioned structures (such as fascin for filopodia shafts or VASP for filopodia tips as well as cortactin and Arp2/3-complex for lamellipodia) in these cultures, which would make such a statement much more reliable.

We performed the requested experiments and included these data in the manuscript (new Fig 1J, Fig EV1H and EV1I). We observed continuous staining with anti-fascin antibodies in Gas2L1-induced filopodia-like protrusions and a more sparse, punctate staining with antibodies against Arp2/3 and cortactin in lamellipodia-like structures.

3) Figure 2 looks very clear, but I was confused by a mixup I think in the images displayed in Supplementary Fig. S2 as compared to both text and legends, since S2E is labeled as control experiment showing that in the absence of Gas2L1, there is no co-alignment of MT and actin bundles. However, the image in S2E clearly shows the presence of bundles and G2L1, but also EB3, so I think the authors have erroneously swapped the images shown in S2E with what's coming next in S2F-H. This must be corrected!

We thank the reviewer for pointing out this mistake, which we have corrected in the revised manuscript.

4) In Figs 5 and 6, the authors quite systematically analyze combined effects of treatments with different concentrations of LatB or jasplakinolide and G2L1 knockdown or overexpression etc, which is interesting and informative, of course, to certain extent, but also quite indirect!

I guess I would have expected a few more of the type of experiments shown in Figure 4, asking much simpler questions, as for instance: How does overexpression or knockdown of G2L1 affect actin dynamics in the axon and in growth cones? In Fig. 4C, the authors show an interesting buckling of MT in growth cones, and perhaps increased growth cone area using lifeact expression. Can the authors expand on this? For instance, show some informative video microscopy data, and provide some quantification on this?

We have added a quantification of the growth cone area (Fig 1G) and of the microtubule configuration inside growth cones (Fig 4D and E) of control, Gas2L1 knockdown and Gas2L1 overexpressing neurons to the manuscript.

Unfortunately, LifeAct expression is not a suitable tool to measure actin dynamics in live neurons by FRAP: the dynamics measured are those of LifeAct-binding to actin rather than of actin itself. Moreover, LifeAct shows selective binding to specific populations of actin filaments and has mild actin-stabilizing effects [6,7], and thus could skew our conclusions. As an alternative approach, we evaluated the potential of using a photoactivatable variant of GFP-actin, but unfortunately this fusion protein does not incorporate into all Gas2L1-positive actin structures (Figure for Reviewers 3).

Figure for Reviewers 3

Still images taken from live-imaging experiments in DIV4 neurons expressing paGFP-Actin and RFP-G2L1 after photoactivation with blue light.

Data information:
Scale bar: 5 μ m

Finally, can the authors explore proposed differential F-actin stabilities in different conditions of GasL1 manipulation by performing a photoactivation activation experiment of actin in different subcellular regions (as for instance in the axon), in analogy to what they have done with tubulin in Fig. 6B? Such experiments would be much more direct than the combinations of inhibitor treatments mentioned above, which could then perhaps be shifted to the Supplement and be replaced by more direct cell biological expts as suggested.

We agree that this would be a good experiment and we performed pilot experiments prompted by the reviewer, but unfortunately we saw that paGFP-actin is not incorporated into all Gas2L1-positive actin structures (see Figure to Reviewers 3).

One more point: I can't really discern the claimed displacement of EB1 upon overexpression of G2L1 in Figure 6A, or the mentioned re-targeting of EB1 to actin. This has to be demonstrated more clearly by counter-stainings with MTs and the actin system, using individual channels for each.

Unfortunately, co-staining of actin and microtubule plus-end tracking proteins such as EBs is difficult. Microtubule plus-end tracking markers can only be visualised using methanol-based fixations, which do not preserve actin structures well and completely prevent labelling with phalloidin. On the other hand, fixations typically used for actin staining (e.g. paraformaldehyde) do not preserve the localization of microtubule plus-end-tracking proteins.

We therefore cannot combine these stainings in the same neurons, and therefore instead indicate that since Gas2L1 co-localizes with the actin (phalloidin) staining, and EB1 staining co-localizes with Gas2L1 upon Gas2L1 overexpression, EB1 must be recruited to phalloidin-labelled actin structures in neurons overexpressing Gas2L1. As a control, we included Gas2L1-SxAA, a mutant deficient in EB binding that therefore does not recruit EB1. We have included clearer figure panels (Fig 6A and B) and rephrased the text to more accurately reflect the data shown.

5) On the positive side, the control experiment with using the actin-binding domain of alpha-actinin to replace the CH-domain in G2L1 is quite interesting, but in the end, I don't quite grasp what the authors mean with: "the localization of Gas2L1 thus appears to be an emergent property of the combination of its specific CH-domain and MT-binding tail)" (last sentence of the Results section)? This is too vague, hence does not mean a lot! Is it really the specific interaction that's relevant or is there anything else?

We have clarified this conclusion in the text: the experiment aimed to show that the behaviour of Gas2L1 is not a simple result of combining any actin-binding domain with the microtubule-binding tail of Gas2L1. Instead, the observed behaviour of Gas2L1 depends on the specific combination of its two domains, suggesting that the interaction between the Gas2L1 CH domain and microtubule-binding tail directs Gas2L1 activity. The statement about emergent properties has been removed.

The problem is that the MT-binding C-terminus used is very large, so how can the authors exclude additional interactions with other proteins to play relevant functions? They should at least mention this! If the authors want to make sure that MT binding alone is what's important, they should narrow

down the MT interaction surface or screen for point mutants to interfere with this much more specifically, which would then help to substantiate such a statement! It might also be worth constructing something like a "mini-G2L1" solely harboring the actin and MT binding surfaces, and then explore the effects of that one, which would demonstrate whether or not additional interactions are really relevant for the specific functions of G2L1 proposed.

We find that the Gas2L1 tail fragment contains at least two microtubule-binding sites, as was also reported for the *Drosophila* Gas2L ortholog Pigs and suggested for Gas2L proteins [3,5]: the GAR domain and the unstructured C-terminal tail. We have now illustrated this in the new Fig. EV3A, where we also show that these two parts of the protein can both promote MT lattice binding. Because of this complexity, making a mini-Gas2L1 is complicated and may not be informative. Further, we fully agree with the reviewer that we cannot exclude that additional interactions with other proteins may contribute, and we have mentioned this in the revised paper in the discussion on p. 13.

Minor comments:

6) I couldn't find Supplementary Figures S3 and S4 until I realized that they were probably not there because each Supplementary Figure fitted the main Figure named accordingly, but this seemed quite strange to me...

This is indeed the numbering for Supplemental figures that we used in the initial submission of our manuscript. It has now been substituted for a more straightforward numbering, where the Supplemental figures are labelled Figure EV1, Figure EV2, in accordance with the Expanded View format of EMBO Reports.

7) On page 14, third para, the text should read ...Gas2L1 likely contributes to a certain level to (not "of") actin stability required for normal growth cone behavior.

We have corrected this sentence.

Referee #3:

Willige et al explore the role and mechanism of action of the actin-microtubule linker Gas2L1 in neurons. The authors combine in vitro reconstitution assays, structure function studies and pull-downs with gain and loss of function studies and pharmacology in primary neuronal culture to further pinpoint the mechanism of action. The authors demonstrate that Gas2L1 promotes the formation of branches along the axon while restricting elongation of the axon in developing hippocampal neurons. They propose that these growth regulatory roles of Gas2L1 occur by stabilising acting a function that requires interaction with the microtubule cytoskeleton and overcoming an autoinhibitory mechanism.

Overall, I like the data and the combination of in vivo with in vitro reconstitution approaches. I find that the authors provide satisfactory data to support most of their arguments and their findings will be of interest to a broad range of reader interested in cytoskeleton regulation in neurons and belong. The paper could improve from few corrections, further discussion and perhaps a couple of extra analysis.

We thank the reviewer for the helpful suggestions.

General comment:

- A key conclusion of the authors is that Gas2L1 regulates neuronal morphology by stabilising the actin cytoskeleton. This conclusion is based on localisation studies, similarity of phenotypes when compared to the effects of actin destabilisation drugs and the rescue with jasplakinolide. However, several microtubule actin linkers have been reported necessary for MT bundle formation, in their

absence, MTs acquired unbundle and misdirected trajectories and fail to extend into growth cone filopodia. To rule out the contribution of Gas2L1 to the regulation of MTs, the authors test the MT stability and the speed of polymerization but do not comment on MT organisation particularly at the growth cone. A set of analysis aiming to describe the bundle organisation of MTs would be very helpful in order to rule out MT regulation.

We have analysed the microtubule growth trajectories and included the data in the new Fig. 4D and 4E of the revised manuscript. Microtubule trajectories in growth cones appear normal in the absence of Gas2L1, reinforcing the idea that it has a function distinct from ACF7/Shot and that it acts primarily on actin rather than microtubules.

- A further conclusion of the authors is that the interaction of Gas2L1 with MTs is able to direct its actin stabilization function. The authors discuss that this property depends on the autoinhibition of Gas2L1 and propose in the discussion that the association of the Gas2L1 tail domain with MTs is expected to liberate the CH domain and to allow Gas2L1 to interact with actin filaments. This comes as a surprise since there are previous discussions suggesting the opposite, as an example please see Pg. 8 "The localization of Gas2L1 to F-actin was also apparent in subcellular areas devoid of MTs. This observation reveals that in contrast to our in vitro experiments, Gas2L1 can localize to actin structures independently of MTs in neurons. This result further strengthens the idea that actin binding may be the first step towards stably relieving the autoinhibition of Gas2L1". This contradiction could require further clarification.

We have clarified this apparent contradiction in the discussion. Presumably, there are additional modes of regulation at play in cells, which are absent from our *in vitro* reconstitutions. Importantly, we find that localization of Gas2L1 to different actin populations in neurons is not sufficient to stabilize them: the stabilization pattern matches the pattern of the microtubule-binding tail fragment, instead of the localization pattern of the full-length protein (Fig. 6I). Therefore, there is no contradiction between the observed localization of Gas2L1 to actin and the statement that its actin-stabilizing properties depend on microtubule binding.

Specific comments:

- Pg. 6 "By contrast, the Tail mutant was able to bind MTs in the absence of actin filaments (Fig. 2F)" --- Tail mutant or Tail domain?

We meant "Tail domain", this was changed in the text.

- Pg.6 "In a composite assay where more actin filaments were available for binding, we observed MTs covered with multiple co-aligned actin filaments (Fig. 2H), whereas no MT-actin co-alignment occurred without Gas2L1 (Fig. S2E)" --- S2E is in the presence of G2L1, do the authors refer to S2E?

We thank the reviewer for pointing out this mistake, which we have corrected in the revised manuscript.

- Pg.8 "...the addition of 10 μ M Latrunculin B (LatB), which inhibits actin polymerization, abolished the interaction between actin and the CH domain (Fig. 3D). This reveals that our pull-down experiments reflect an interaction between Gas2L1 and actin filaments rather than actin monomers" --- Could the authors comment on the finding that full length Gas2L1+ CH domain ability to pull down actin when coexpressed in not abolished with 10 μ M Latrunculin B?

This result is indeed confusing, and with some additional experiments using lysis buffers with varying salt concentrations we found that the ability of full-length Gas2L1 (but not the CH domain) to pull down actin in the presence of LatB strongly changes depending on the ionic strength of the buffer. We decided to avoid making any statements on this difference and removed the figure panel and the corresponding conclusion.

- Fig.S2 Potential levelling mistake: F-H TIRF images (F) (or H??) and kymographs (G, H) (or F, G) showing specific Gas2L1 localization to MT-actin overlaps (E instead of H?) and absence of plus-end tracking in an in vitro reconstitution with Gas2L1, F-actin (1 μ M), MTs and EB3 (is the first

panel in *F tubulin*? If so mark it in accordance). These data indicate that EB3 does not influence the localization of Gas2L1 in this system, even when EB3 tracks growing MT plus ends (H) (or G?).

We erroneously swapped the references to individual panels of Fig. S2 (now EV2) in the text, and we have corrected this in the new version. We use the label "MTs", except when the freshly incorporated tubulin and the MT seed have a different fluorescent label.

- Pg. 9 "... These results are in agreement with our observation that Gas2L1 is not a plus-end tracking protein in vitro. However, Gas2L1 does not recruit EB3 in vitro (Fig. S2H)...." --- S2H or S2G?

We meant S2G (now EV2G), the mistake has been corrected.

- Pg.11 when describing the role of MTs in Gas2L1 activity, the authors compare GFP-Gas2L1-SxAA with GFP-CH. GFP-CH tends to localise to dendrites the position where it protects actin. However GFP-Gas2L1-SxAA also equally localises to dendrites but surprisingly fails to protect dendritic actin and instead protects axonal actin. The authors comment by saying that "... SxAA mutant displayed slightly less efficient axonal localization and actin stabilization (Fig. 6H), implying that the SxIP motif of Gas2L1 may function to enhance the affinity of the protein for axonal MTs" but the stamen does not represent the data presented and instead fails to support the hypothesis. This would need further clarification.

We have adjusted the text to reflect the results more accurately. As explained above, a key observation here is that the localization of wild type full length Gas2L1 does not fully match its actin-stabilising activity probed with LatB: the strongest actin stabilization is observed in axons, whereas the localization of Gas2L1 is not polarised. The SxAA mutant shows the same trend, however, it is less abundant in the axon and more enriched in non-axonal neurites. This is likely due to the fact that this mutant has a somewhat lower overall affinity for microtubules because it does not interact with the microtubule-binding EB proteins. Importantly, the tail part of Gas2L1-SxAA can still bind to microtubules and to the CH-domain of Gas2L1 (Fig. 3A, Fig. EV3A; the latter figure also illustrates the effect of the SxAA mutation on MT binding of the Gas2L1 Tail domain) and thus its actin stabilization pattern is similar to that of the wild type protein.

References

1. Krendel M, Zenke FT, Bokoch GM (2002) Nucleotide exchange factor GEF-H1 mediates cross-talk between microtubules and the actin cytoskeleton. *Nat Cell Biol* **4**: 294-301
2. Au FK, Jia Y, Jiang K, Grigoriev I, Hau BK, Shen Y, Du S, Akhmanova A, Qi RZ (2017) GAS2L1 Is a Centriole-Associated Protein Required for Centrosome Dynamics and Disjunction. *Dev Cell* **40**: 81-94
3. Goriounov D, Leung CL, Liem RK (2003) Protein products of human Gas2-related genes on chromosomes 17 and 22 (hGAR17 and hGAR22) associate with both microfilaments and microtubules. *J Cell Sci* **116**: 1045-1058
4. Pines MK, Housden BE, Bernard F, Bray SJ, Roper K (2010) The cytolinker Pigs is a direct target and a negative regulator of Notch signalling. *Development* **137**: 913-922
5. Girdler GC, Applewhite DA, Perry WM, Rogers SL, Roper K (2016) The Gas2 family protein Pigs is a microtubule +TIP that affects cytoskeleton organisation. *J Cell Sci* **129**: 121-134
6. Belin BJ, Goins LM, Mullins RD (2014) Comparative analysis of tools for live cell imaging of actin network architecture. *Bioarchitecture*, **4**: 189-202
7. Spracklen AJ, Fagan TN, Lovander KE, Tootle TL (2014) The pros and cons of common actin labeling tools for visualizing actin dynamics during *Drosophila* oogenesis. *Dev Biol.* **393**:209-226

2nd Editorial Decision

5 August 2019

Thank you for submitting the revised version of your manuscript. It has now been seen by all of the original referees.

As you can see, all referees find that the study is significantly improved during revision and recommend publication. Before I can accept the manuscript, I need you to address some editorial points below:

REFeree REPORTS

Referee #1:

Revisions are adequate.

Referee #2:

The authors have addressed all major points raised in my previous review to my satisfaction, so in my view, the manuscript can now be published as is.

Referee #3:

I am satisfied with the revised manuscript

2nd Revision - authors' response

7 August 2019

The authors performed all minor editorial changes.

3rd Editorial Decision

15 August 2019

Thank you for submitting your revised manuscript. I have now taken a look at everything and all looks fine. Therefore I am very pleased to accept your manuscript for publication in EMBO Reports.

Corresponding Author Name: Casper Hoogenraad, Gijssje Koenderink, Anna Akhmanova

Manuscript Number: EMBOR-2019-47732